# Can Language Models Solve Graph Problems in Natural Language?

**Heng Wang**[* 1], **Shangbin Feng**[* 2], **Tianxing He**[2], **Zhaoxuan Tan**[3], **Xiaochuang Han**[2], **Yulia Tsvetkov**[2]
[1]Xi'an Jiaotong University   [2]University of Washington   [3]University of Notre Dame
wh2213210554@stu.xjtu.edu.cn, shangbin@cs.washington.edu

## Abstract

Large language models (LLMs) are increasingly adopted for a variety of tasks with *implicit graphical structures*, such as planning in robotics, multi-hop question answering or knowledge probing, structured commonsense reasoning, and more. While LLMs have advanced the state-of-the-art on these tasks with structure implications, whether LLMs could explicitly process textual descriptions of graphs and structures, map them to grounded conceptual spaces, and perform structured operations remains underexplored. To this end, we propose NLGraph (Natural Language Graph), a comprehensive benchmark of graph-based problem solving designed in natural language. NLGraph contains 29,370 problems, covering eight graph reasoning tasks with varying complexity from simple tasks such as connectivity and shortest path up to complex problems such as maximum flow and simulating graph neural networks. We evaluate LLMs (GPT-3/4) with various prompting approaches on the NLGraph benchmark and find that 1) language models *do* demonstrate preliminary graph reasoning abilities, 2) the benefit of advanced prompting and in-context learning diminishes on more complex graph problems, while 3) LLMs are also (un)surprisingly brittle in the face of spurious correlations in graph and problem settings. We then propose Build-a-Graph Prompting and Algorithmic Prompting, two instruction-based approaches to enhance LLMs in solving natural language graph problems. Build-a-Graph and Algorithmic prompting improve the performance of LLMs on NLGraph by 3.07% to 16.85% across multiple tasks and settings, while how to solve the most complicated graph reasoning tasks in our setup with language models remains an open research question. The NLGraph benchmark and evaluation code are available at https://github.com/Arthur-Heng/NLGraph.

## 1 Introduction

Originally designed for textual data, large language models (LLMs) are increasingly leveraged for tasks beyond language processing. In robotics and planning, LLMs are adopted to guide agents through structured environments [Huang et al., 2022, Andreas, 2022]. In theory-of-mind reasoning, LLMs are required to maintain and update local and global graphs that reflect the beliefs of different characters [Adhikari et al., 2020, Ammanabrolu and Riedl, 2021]. In structured commonsense reasoning, LLMs are expected to generate graph-based action plans to achieve objectives with diversified prerequisites [Tandon et al., 2019, Madaan et al., 2022]. In multi-hop question answering, LLMs implicitly find connections and paths among a vast network of entities and concepts [Creswell et al., 2023]. Together these works demonstrate that LLMs are widely adopted for tasks with *implicit graphical structures* while achieving preliminary success. However, one underlying yet crucial question remains underexplored: *Can LLMs reason with graphs?* More concretely, *are LLMs capable of mapping textual descriptions of graphs and structures to grounded conceptual spaces and solving*

---

[*]equal contribution

*graph algorithm problems explicitly with natural language?* The answer to this question has profound implications for large language model applications with implicit graphs and structures, the reasoning ability of LLMs in advanced and graph-based settings, and more.

To this end, we propose the Natural Language Graph (NLGraph) benchmark, a comprehensive testbed of graph and structured reasoning designed for language models and in natural language. NLGraph contains a total of 29,370 problems, covering eight graph reasoning tasks with varying complexity from intuitively simple tasks such as *connectivity*, *cycle*, and *shortest path* to more complex problems such as *topological sort*, *maximum flow*, *bipartite graph matching*, *Hamilton path*, and *simulating graph neural networks*. We control for problem difficulty through generated graph size, network sparsity, numeric range, and more, presenting easy, medium, and hard subsets in each distinct graph reasoning task to enable fine-grained analysis. In addition to using exact match accuracy as a standard metric, we also design several partial credit solutions for various graph reasoning tasks.

With the NLGraph benchmark, we evaluate whether various large language models [Brown et al., 2020, Ouyang et al., 2022, Bubeck et al., 2023] could perform graph-based reasoning and whether different prompting techniques [Brown et al., 2020, Wei et al., 2022, Kojima et al., 2022, Zhou et al., 2023, Wang et al., 2023] improve the graph reasoning abilities of large language models. Extensive experiments on the NLGraph benchmark demonstrate that:

1. **LLMs *do* possess preliminary graph reasoning abilities.** Specifically, large language models demonstrate an impressive level of performance that is 37.33% to 57.82% above the random baseline on simple graph reasoning tasks such as connectivity, cycle, and shortest path. With chain-of-thought prompting, LLMs could generate intermediate steps that are sound and accurate while further improving task performance.

2. **The benefit of advanced prompting methods diminishes with complex problems.** On one hand, chain-of-thought [Wei et al., 2022], least-to-most [Zhou et al., 2023], and self-consistency [Wang et al., 2023] successfully enhance the graph reasoning abilities of LLMs on simple tasks such as cycle and shortest path. On the other hand, these approaches are mostly ineffective, even counterproductive in certain settings, on more complex graph reasoning problems such as topological sort and Hamilton path.

3. **Learning from examples did not happen on complex graph reasoning problems.** While in-context learning is widely credited for teaching LLMs to learn from examples [Brown et al., 2020], its benefit on more advanced graph reasoning tasks is unclear: Few-shot in-context learning fails to improve over zero-shot prompting across multiple tasks, while increasing the number of exemplars may even be counterproductive for tasks such as Hamilton path.

4. **LLMs are (un)surprisingly brittle to spurious correlations in problem settings.** We find that in two special cases in the connectivity task (chain and clique), LLMs perform much worse than the general dataset with a performance drop of more than 40% across various settings. This indicates that large language models are implicitly relying on certain spurious correlations (for example, use node mention frequency to determine connectivity), falling short of performing robust structured reasoning in graph-based contexts.

To improve large language models as better graph reasoners, we propose two instruction-based prompting approaches to better elicit the graph reasoning abilities of large language models. *Build-a-Graph prompting* encourages LLMs to map the textual descriptions of graphs and structures to grounded conceptual spaces [Patel and Pavlick, 2022] before tackling the specific problem through a one-sentence instruction, while *Algorithmic prompting* instructs LLMs to revisit the algorithmic steps for a given task before learning from in-context exemplars. Experiments demonstrate that build-a-graph and algorithmic prompting successfully empower LLMs to better tackle graph reasoning problems, resulting in 3.07% to 16.85% performance gains across multiple tasks, while the most complicated graph reasoning problems remain an open research question.

## 2 The NLGraph Benchmark

To examine whether language models are capable of reasoning with graphs and structures, we curate and propose the Natural Language Graph (NLGraph) benchmark. Specifically, we first employ a random graph generator to generate graphs and structures while controlling for the network size, graph sparsity, and more. We then adopt the generated graphs as bases to synthetically generate problems for eight graph-based reasoning tasks with varying algorithmic difficulties. We control for

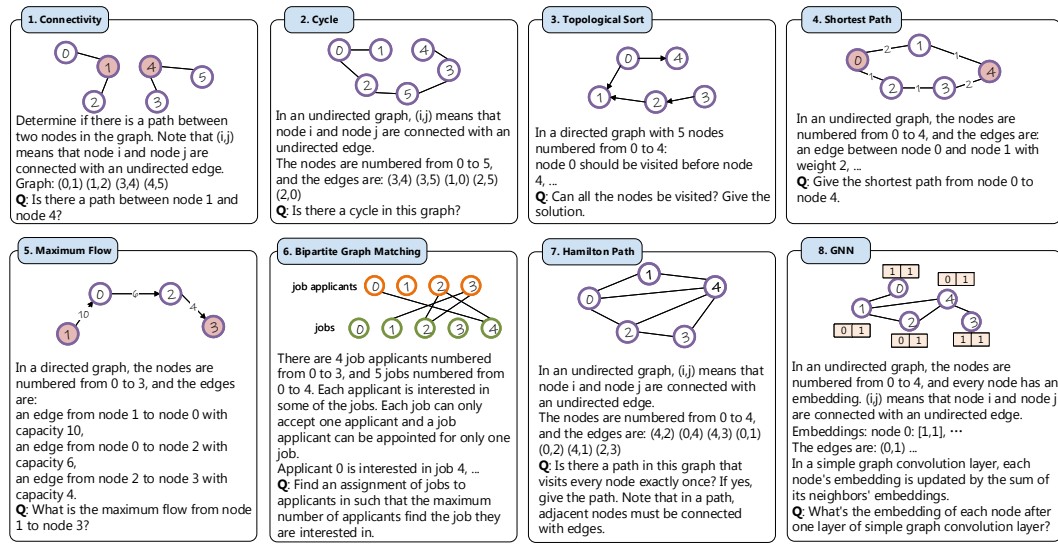

Figure 1: Overview of the NLGraph Benchmark, featuring eight tasks with varying complexity. We show an intuitive figure representing each task along with the example natural language prompts being passed to the LLMs.

problem difficulty within each of the eight tasks, resulting in easy, medium, and hard subsets in each graph reasoning task to enable difficulty scaling and fine-grained analysis.

## 2.1 Random Graph Generator

We first employ a general-purpose random graph generator to generate base graphs while using the number of nodes and graph density to control for complexity. Formally, to generate a graph $\mathcal{G} = \{\mathcal{V}, \mathcal{E}\}$ where $\mathcal{V}$ and $\mathcal{E}$ denote the set of nodes and edges, we specify the number of nodes $n$, thus $\mathcal{V} = \{v_1, v_2, \ldots, v_n\}$, and $|\mathcal{V}| = n$. We then specify the edge probability $p$ such that all the edges are generated according to $P(e_{ij} \in \mathcal{E}) = p$, where $e_{ij}$ is an edge from $v_i$ to $v_j$. The edges could be directed or undirected depending on the task. We use varying $n$ and $p$ values to control for the complexity of the random graph structure. Building on top of these generated base graphs, we also adopt graph edits and other difficulty control factors tailored for each task.

## 2.2 Tasks

Armed with the general-purpose random graph generator, we adapt the synthetically generated graphs to eight graph reasoning tasks with varying complexity and describe the problem setups in natural language. Specifically, we first generate easy, medium, and hard subsets of base graphs for each task by controlling for node amount and graph sparsity. We then adapt or edit the base graphs for each task and design queries accordingly. We present an overview of the NLGraph benchmark with specific natural language instructions in Figure 1.

- **Task 1: Connectivity** In an undirected graph $\mathcal{G} = \{\mathcal{V}, \mathcal{E}\}$, two nodes $u$ and $v$ are *connected* if there exists a sequence of edges from node $u$ to node $v$ in $\mathcal{E}$. We randomly select two nodes in the base graphs $u, v \in \mathcal{V}$ to ask whether node $u$ and node $v$ are connected with a true/false question. We retain a balanced set of questions where half of the node pairs are connected and the other half are not connected by discarding additional questions.

- **Task 2: Cycle** In an undirected graph $\mathcal{G} = \{\mathcal{V}, \mathcal{E}\}$, a cycle is a non-empty trail $(e_1, e_2, \ldots, e_n)$ with a node sequence $(v_1, v_2, \ldots, v_n, v_1)$. We present an undirected graph $\mathcal{G} = \{\mathcal{V}, \mathcal{E}\}$ to ask whether there exists a cycle through true/false questions. We retain base graphs without cycles as the False subset, while we randomly add edges to these base graphs to generate graphs with cycles as the True subset. We retain a balanced set of cyclic and noncyclic graphs in the dataset.

- **Task 3: Topological Sort** A topological sort of a directed graph is a linear ordering of its nodes such that for every directed edge $(u, v)$ from node $u$ to node $v$, $u$ comes before $v$ in the ordering.

| Subset | Connect. | Cycle | Topo. Sort | Shortest Path | Max. Flow | Bipartite Graph | Hamilton Path | GNNs |
|---|---|---|---|---|---|---|---|---|
| # EASY | 352 / 730 | 150 / 300 | 180 / 360 | 180 / 360 | 150 / 300 | 300 / 600 | 150 / 300 | 100 / 200 |
| SPEC. | $n$: 5-10 | $n$: 5-10 | $n$: 5-10 | $n$: 5-10 | $n$: 5-10 | $n$: 6-20 | $n$: 5-10 | $n$: 5-8 |
| # MEDIUM | 1,200 / 8,580 | 600 / 1,800 | 150 / 1,350 | / | / | / | / | / |
| SPEC. | $n$: 11-25 | $n$: 11-25 | $n$: 11-25 | / | / | / | / | / |
| # HARD | 680 / 7,090 | 400 / 2,000 | 200 / 1,200 | 200 / 1,200 | 200 / 1,200 | 210 / 1,260 | 200 / 600 | 140 / 840 |
| SPEC. | $n$: 26-35 | $n$: 26-35 | $n$: 26-35 | $n$: 11-20 | $n$: 11-20 | $n$: 17-33 | $n$: 11-20 | $n$: 9-15 |

Table 1: Statistics of the NLGraph benchmark. We use A / B to indicate that there are A and B problems in the standard and extended set of NLGraph. SPEC. denotes difficulty specifications.

The task is to find a valid topological sort given a directed graph and there could be multiple valid solutions. We ask LLMs to generate a valid topological sort for the given directed graph and employ an external program to examine its correctness.

- **Task 4: Shortest Path** The shortest path between two nodes is the path with the sum of edge weights minimized. Given an undirected graph $\mathcal{G} = \{\mathcal{V}, \mathcal{E}\}$, a positive weight $w$ for each edge, and two nodes $u$ and $v$, the task is to find the shortest path between node $u$ and node $v$ and its corresponding path length. We filter the generated base graphs by specifying that the number of nodes on the correct shortest path is as least $\ell$, where $\ell$ is chosen from $\ell_{min}$ to $\ell_{max}$ for each question to sever as an additional difficulty control measure. We adopt two metrics: *exact match*, where the LLM solution is a valid path and optimal, and *partial credit*, where the LLM solution is valid and is the $\mathrm{rank}_i$-th shortest among all the possible paths (i.e. the number of shorter paths plus one). The partial credit score for each problem can be formulated as $\mathrm{PC} = 1/\mathrm{rank}_i$.

- **Task 5: Maximum Flow** Let $\mathcal{G} = \{\mathcal{V}, \mathcal{E}\}$ be a network with two nodes $s, t \in \mathcal{V}$ being the source and the sink. Each edge is associated with a capacity $c$, the maximum amount of flow that can pass through the edge. We ask LLMs to generate a plan to route as much flow as possible from the source to the sink. We evaluate in both exact match with the optimal plan and partial credit for this task, where partial credit can be formulated as $\mathrm{PC} = \begin{cases} t/s, & \text{if } t \leq s \\ 0, & \text{if } t > s \end{cases}$, where $s$ is the flow value under the optimal plan, and $t$ is the flow value of the solution generated by LLMs.

- **Task 6: Bipartite Graph Matching** In an undirected graph $\mathcal{G} = \{\mathcal{V}, \mathcal{E}\}$, a matching is a set of edges without common nodes. A bipartite graph is a graph whose nodes can be divided into two disjoint sets $\mathbf{U}$ and $\mathbf{V}$, and in each set no nodes are adjacent to each other. Given a bipartite graph, the task is to find the matching that maximizes the number of edges. We use an external program to evaluate whether the solution generated by LLMs is valid and optimal.

- **Task 7: Hamilton Path** In an undirected graph, a Hamilton path is a path that visits every node exactly once. Given an undirected graph $\mathcal{G} = \{\mathcal{V}, \mathcal{E}\}$, the task is to find a valid Hamilton path. We filter generated base graphs to ensure that at least one valid Hamilton path exists and use an external program to evaluate the LLM solution.

- **Task 8: Graph Neural Networks** Given an undirected graph $\mathcal{G} = \{\mathcal{V}, \mathcal{E}\}$, and a two-dimension node embedding $\mathbf{x}_i$ for each node, the task is to perform $\ell$ layers of message passing, *i.e.* to update the node embedding with the sum of all the neighbors' embeddings. Formally, $\mathbf{x}_i^{(\ell+1)} = \sum_{j \in \mathcal{N}_i} \mathbf{x}_j^{(\ell)}$ where $\mathcal{N}_i$ denotes the neighborhood of node $i$ and $(\ell)$ denotes the $\ell$-th layer. We use an exact match with the correct node embeddings and two types of partial credits for this task. Specifically, the first partial credit is the percentage of the nodes whose embedding is correct (PC), and the second is the average of all the values' relative errors for the standard answer (RE). The relative error is formulated as $RE = \frac{|x-y|}{\max(x,y)}$, where $x$ is the value generated by LLMs and $y$ is the value in the standard answer, averaged across all embedding dimensions.

## 2.3 Benchmark Statistics

Using the above methodology, we generate the NLGraph benchmark with 5,902 problems in a standard version and 29,370 problems in an extended version. Intuitively easier tasks, including connectivity, cycle, and topological sort problems, are further divided into easy, medium, and hard subsets based on graph size, sparsity, among other difficulty control factors. More algorithmically advanced tasks, including shortest path, maximum flow, bipartite graph matching, Hamilton path,

| Method | Connectivity | | | | Cycle | | | | Shortest Path | | | | |
|---|---|---|---|---|---|---|---|---|---|---|---|---|---|
| | Easy | Medium | Hard | Avg. | Easy | Medium | Hard | Avg. | Easy | Hard | Easy (PC) | Hard (PC) | Avg. |
| RANDOM | 50.00 | 50.00 | 50.00 | 50.00 | 50.00 | 50.00 | 50.00 | 50.00 | 6.07 | 6.69 | 14.73 | 13.81 | 17.81 |
| ZERO-SHOT | 83.81 | 72.75 | 63.38 | 71.31 | 50.00 | 50.00 | 50.00 | 50.00 | 29.40 | 21.00 | 46.00 | 26.76 | 30.79 |
| FEW-SHOT | 93.75 | 83.83 | 76.61 | 84.73 | 80.00 | **70.00** | **61.00** | **70.33** | 31.11 | 26.00 | 49.19 | 35.73 | 35.51 |
| CoT | **94.32** | 82.17 | 77.21 | 84.57 | **84.67** | 63.33 | 53.25 | 66.75 | 63.89 | **29.50** | 76.84 | 35.79 | 51.51 |
| 0-CoT | 79.55 | 65.83 | 68.53 | 71.30 | 55.33 | 57.67 | 49.00 | 54.00 | 8.89 | 7.50 | 62.39 | **43.95** | 32.03 |
| CoT+SC | 93.18 | **84.50** | **82.79** | **86.82** | 82.00 | 63.67 | 53.50 | 66.39 | **68.89** | 29.00 | **80.25** | 38.47 | **54.15** |

Table 2: Model performance on the connectivity, cycle, and shortest path tasks. PC denotes partial credit. Large language models with CoT or CoT+SC prompting greatly outperforms the random baseline by 37.33% to 57.82%, indicating that LLMs have preliminary graph reasoning abilities.

and graph neural networks problems, are divided into easy and hard subsets. We present benchmark statistics in Table 1. Accuracy (whether the true/false answer is correct, whether the proposed solution is valid) is the default evaluation metric, while tasks 4, task 5, and task 8 have additional partial-credit metrics aforementioned. We envision NLGraph as a comprehensive testbed for graph and structured reasoning for large language models.

# 3 Experimental Settings

Based on the NLGraph benchmark, we aim to investigate *whether language models can solve graph algorithm problems in natural language* by evaluating large language models and different prompting approaches.

**Baselines** We adopt a wide range of prompting approaches as baselines. Specifically, zero-shot prompting (ZERO-SHOT), few-shot in-context learning (FEW-SHOT) [Brown et al., 2020], chain-of-thought prompting (CoT) [Wei et al., 2022], zero-shot chain-of-thought (0-CoT) [Kojima et al., 2022], least-to-most (LTM) [Zhou et al., 2023], and self-consistency (SC) [Wang et al., 2023] are leveraged to tackle various graph reasoning tasks in the NLGraph benchmark.

We also adopt a RANDOM baseline: for true/false questions such as connectivity and cycle, we use RANDOM to denote a baseline that randomly selects an answer from true and false with an expected accuracy of 50%; For the shortest path task, RANDOM denotes a baseline that randomly selects a valid path between the query node pair. For the maximum flow task, RANDOM denotes a baseline that randomly selects a value between 0 and the sum of all the edges' capacities. The performance comparison between different prompting techniques and the RANDOM baseline could indicate whether LLMs are capable of performing graph reasoning instead of giving randomly generated answers.

**Models and Settings** We use TEXT-DAVINCI-003 as the default large language model, while we also evaluate three additional LLMs (GPT-3.5-TURBO, CODE-DAVINCI-002, and GPT-4), presenting results in Appendix E. For all baselines except self-consistency, we set temperature $\tau = 0$; For self-consistency prompting, we sample five chain-of-thought responses with temperature $\tau = 0.7$. For few-shot prompting techniques (i.e., FEW-SHOT, CoT, LTM, and CoT+SC), the input prompt includes $k$ exemplars selected from the extended version before the problem of interest. For the connectivity task and cycle task, we set $k$ to 4, for the GNN task, we set $k$ to 1 due to the context size limit, while for other tasks $k$ is 5. We evaluate LLMs and various prompting techniques on the standard set of NLGraph due to monetary costs, while we encourage future research to leverage the extended version for enhanced evaluation.

# 4 Results

## 4.1 LLMs Have (Preliminary) Graph Reasoning Abilities

We first find that on intuitively simple graph reasoning tasks, large language models achieve impressive performance and demonstrate preliminary graph reasoning abilities. As demonstrated in Table 2, LLM performance on the connectivity, cycle, and shortest path tasks is significantly better than the RANDOM baseline, indicating that LLMs are not giving random answers and they *do* have preliminary graph reasoning abilities. In the first two tasks with true/false questions, LLM performance is 37.33% to 57.82% higher than random with CoT or CoT+SC prompting. The performance is also impressive

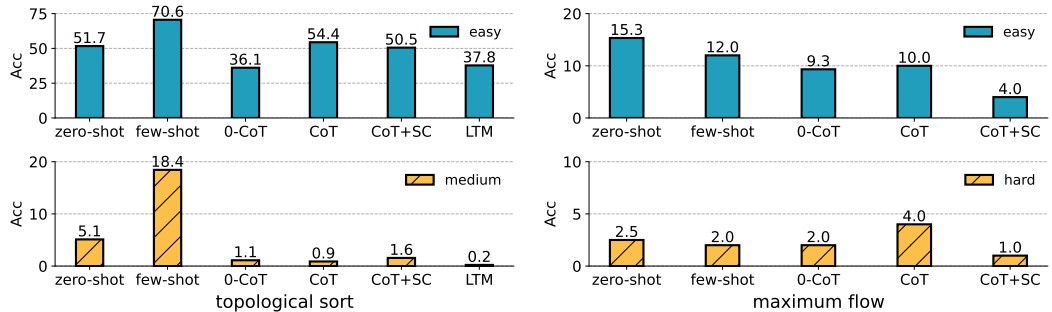

Figure 2: (left) Model performance on the topological sort task. CoT, LTM, and self-consistency are mostly ineffective on this problem. (right) Model performance on the maximum flow task. FEW-SHOT prompting outperforms CoT+SC prompting on both easy and hard subsets, suggesting that LLMs fall short of generating valid intermediate steps to solve the more complex graph reasoning problem. Together these results demonstrate that advanced prompting is ineffective for advanced graph reasoning.

in the ZERO-SHOT setting, as the accuracy is 33.81% and 23.33% higher on the connectivity and shortest path task than random even without any exemplars and chain-of-thought reasoning. Though the shortest path is intuitively harder than the first two true/false tasks since it requires generating the specific shortest path in the response, LLM combined with CoT and CoT+SC achieves an accuracy 22.81% to 62.83% higher than RANDOM. Together these results demonstrate that large language models *do* possess preliminary abilities to process graphs and structures in input texts.

## 4.2 Mixed Results with Advanced Prompting

Table 2 shows that advanced prompting techniques such as chain-of-thought [Wei et al., 2022] and self-consistency [Wang et al., 2023] successfully improve performance on simple graph reasoning tasks. For the task of simulating graph neural networks (Table 3), chain-of-thought also significantly improves model performance. However, these approaches can be ineffective, even counterproductive on more complex graph reasoning tasks. From the results on the topological sort and maximum flow task (Figure 2), we observe that CoT, CoT+SC, and LTM prompting generally underperform FEW-SHOT prompting. We believe this may be be-

| Method | PC (↑) | ACC (↑) | RE (↓) |
|---|---|---|---|
| ZERO-SHOT | 13.61 | 0.00 | 20.04 |
| FEW-SHOT | 20.04 | 0.00 | 37.83 |
| CoT | **64.55** | **31.00** | 14.34 |
| 0-CoT | 13.85 | 0.00 | 44.55 |
| CoT+SC | 63.92 | 28.00 | **13.28** |

Table 3: Model performance on the task of simulating graph neural networks. PC and RE are two partial credit metrics introduced in §2.2. Chain-of-thought prompting significantly improves the model performance across all metrics.

cause LLMs fail to learn the correct way to generate intermediate steps in the face of complex graph reasoning tasks. This casts doubt on the generalizability of CoT, LTM, and self-consistency to more advanced graph reasoning problems.

## 4.3 In-Context Learning Can be Counterproductive

Although in-context learning is widely attributed to teaching LLMs to learn from in-context exemplars [Brown et al., 2020], we observe that few-shot in-context learning fails to improve LLM performance over complex graph reasoning problems. For tasks such as Hamilton path and bipartite graph matching (Figure 3), zero-shot prompting generally outperforms few-shot learning with in-context exemplars. The results suggest that LLMs fail to learn from the in-context exemplars when the problem involves hundreds of graph-based reasoning steps. In-context exemplars might even distract large language models, evident in that few-shot learning underperforms zero-shot learning by 1.00% to 10.48% on Hamilton path and bipartite graph matching. We further study the correlation between the number of exemplars and model performance on the Hamilton path task. As illustrated in Figure 4, when the number of exemplars increases, model performance is not trending up, and the performance is consistently lower than zero-shot prompting. These results further suggest that in-context learning could be counterproductive in complex structured reasoning.

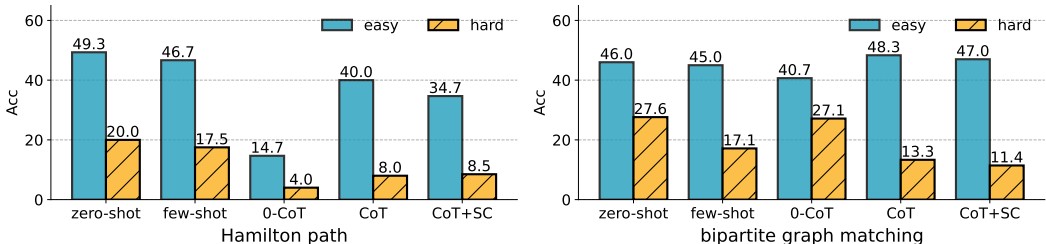

Figure 3: (left) Model performance on the Hamilton path task. ZERO-SHOT prompting consistently outperforms all other prompting techniques. (right) Model performance on the bipartite graph matching task. The effect of in-context learning and advanced prompting is also mostly marginal in this complex graph reasoning problem. Together these results demonstrate that in-context learning can be counterproductive in advanced graph reasoning problems.

| Dataset | ZERO-SHOT | FEW-SHOT | CoT | 0-CoT | CoT+SC | Avg. |
|---|---|---|---|---|---|---|
| GENERAL | 74.67 | 83.33 | 85.33 | 66.00 | 82.67 | 78.40 |
| CHAIN | 51.67 (-23.00) | 45.00 (-35.33) | 40.83 (-44.50) | 92.50 (+26.50) | 44.17 (-38.50) | 54.83 (-23.57) |
| CLIQUE | 60.83 (-13.84) | 73.33 (-10.00) | 85.00 (-0.33) | 52.50 (-13.50) | 83.33 (+0.66) | 71.00 (-7.40) |

Table 4: Model performance on the chain and clique subset of the connectivity task. Large language models indeed rely on spurious correlations in problem settings, evident in the reduced performance on the two special cases compared to the general connectivity task.

## 4.4 LLMs are (Un)surprisingly Brittle

Although large language models achieved performance that is significantly better than random on simple graph reasoning tasks (§4.1), we hypothesize that LLMs may be able to reach the correct answer by leveraging certain spurious correlations. For example, on the connectivity task, since nodes with higher degrees are more frequently mentioned and they are more likely to be connected, LLMs might just be counting node mentions instead of actually finding paths. To this end, we design two special cases (chain and clique) for the connectivity task that are the exact opposite of the spurious correlation.

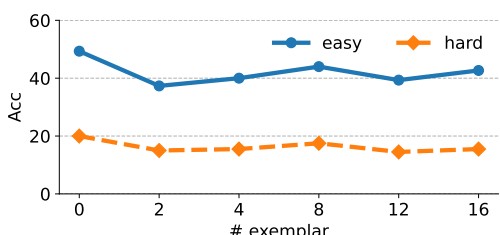

Figure 4: Model performance on the Hamilton path task with an increasing amount of exemplars. When more in-context exemplars are introduced, model performance is not trending up and is consistently lower than zero-shot prompting in both difficulty settings.

**Chain** We firstly generate a graph with $k$ components, where each components is a chain. Formally, $\mathcal{G} = \{\mathcal{V}, \mathcal{E}\}$ can be divided into subgraphs $\mathcal{G}_1, \mathcal{G}_2, \ldots, \mathcal{G}_k$, where subgraph $\mathcal{G}_i$ consists of a chain of nodes and edges $v_{i1}, e_{i1}, v_{i2}, e_{i2}, \ldots, e_{it_i}, v_{it_i}$. We then randomly select query node pairs that are at the two ends of a chain, *i.e.* $v_{i1}$ and $v_{it_i}$. These nodes only have a degree of one but they are actually connected through the chain structure. We curate a chain dataset with 120 examples.

**Clique** We first generate a graph with $k$ densely connected subgraphs. Formally, $\mathcal{G} = \{\mathcal{V}, \mathcal{E}\}$ can be divided into subgraphs $\mathcal{G}_1, \mathcal{G}_2, \ldots, \mathcal{G}_k$, where subgraph $\mathcal{G}_i$ is generated by the random graph generator (§2.1) with a high edge probability $p \in \{0.7, 1.0\}$. We then randomly select query node pairs with each pair in two different densely connected subgraphs, $\mathcal{G}_i$ and $\mathcal{G}_j$ ($i \neq j$). These nodes are associated with a large number of edges and thus are frequently mentioned in the natural language prompt, but they belong to two different subgraphs and are not connected. We curate a clique dataset with 120 examples.

We evaluate LLMs with different prompting approaches on the two special cases and compare performances with the general connectivity task in Table 4. LLM performs much worse than ordinary

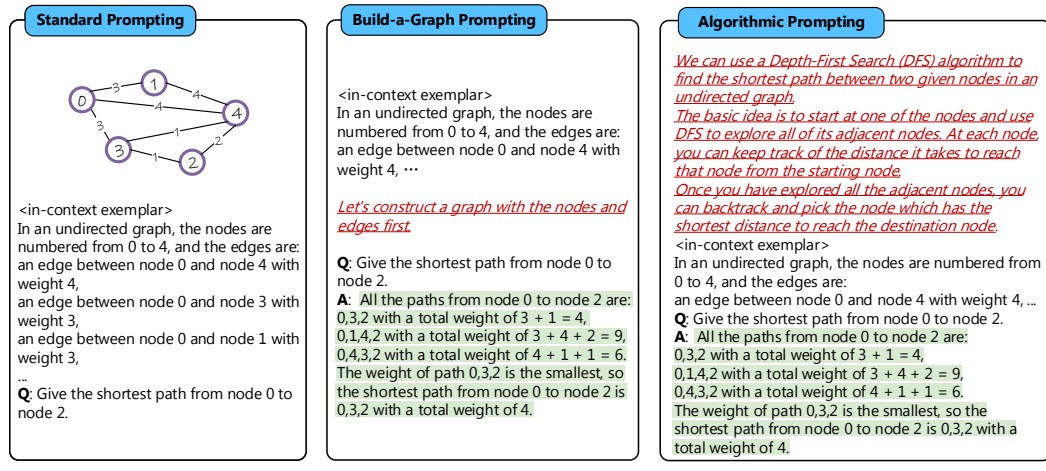

Figure 5: Overview of Build-a-Graph prompting and Algorithmic prompting, aiming to make LLMs better at reasoning with graphs by encouraging conceptual grounding and algorithmic reflection. Red underline indicates our proposed instructions while green background indicates LLMs' generation.

cases with a performance drop of more than 40% across various settings. This suggests that LLMs are indeed (un)surprisingly vulnerable to spurious correlations in structured reasoning.

## 5 Making Language Models Better Graph Reasoners

### 5.1 Methodology

We propose two instruction-based prompting techniques that improve the graph reasoning ability of LLMs, which can be used together with in-context learning and chain-of-thought prompting.

**Build-a-Graph Prompting (BAG)** We hypothesize that it might be helpful to map the textual descriptions of graphs to grounded conceptual spaces [Patel et al., 2021] before tackling the specific problem, *i.e.* visualizing the graph structure on an implicit mental sketchpad. To this end, we propose to use instruction-based prompting by appending "*Let's construct a graph with the nodes and edges first.*" after the textual description of the graph is explicitly given. We envision this straightforward instruction would provide LLMs with the buffer zone to digest graph information, map them to grounded conceptual spaces, and better prepare for the incoming query.

**Algorithmic Prompting** We hypothesize that in order to generate sound and accurate solutions, LLMs would benefit from revisiting and reciting the relevant algorithm for a given task [Sun et al., 2023]. To this end, we propose to first prepend the algorithm details before the in-context examples by adding "We can use a Depth-First Search (DFS) algorithm . . ." for the shortest path task. We similarly provide the algorithm descriptions of other graph reasoning tasks in their respective prompts. We envision algorithmic prompting as empowering LLMs with a general understanding of how to solve the problem before actually solving it.

We illustrate our proposed build-a-graph prompting and algorithmic prompting in Figure 5. In the following, we use these two approaches together with chain-of-thought prompting and evaluate on three representative tasks in the NLGraph benchmark with varying difficulties.

### 5.2 Results

Table 5 shows the results of our proposed prompting methods on three representative tasks with varying complexity. On the two easier tasks, cycle and shortest path, our instruction-based prompting methods resulted in a 3.07% to 16.85% performance gain across the two tasks. However, on the more complex task, Hamilton path, the two natural language instructions are largely ineffective. The result suggests that the two simple instruction-based prompting techniques successfully improve LLMs' graph reasoning abilities on relatively easy graph reasoning tasks, while how to make LLMs better

| Method | Cycle | | | | Shortest Path | | | | | Hamilton Path | | |
|---|---|---|---|---|---|---|---|---|---|---|---|---|
| | Easy | Medium | Hard | Avg. | Easy | Hard | Easy (PC) | Hard (PC) | Avg. | Easy | Hard | Avg. |
| CoT | 84.67 | 63.33 | 53.25 | 66.75 | 63.89 | 29.50 | 76.84 | 35.79 | 51.51 | **40.00** | **8.00** | **24.00** |
| CoT+BAG | **86.00** | 69.33 | 62.00 | **72.44** | **67.78** | **33.50** | **79.20** | **42.56** | **55.76** | 38.67 | 6.00 | 22.34 |
| CoT+ALGORITHM | 77.33 | **74.00** | **64.00** | 71.78 | 63.89 | 28.00 | 76.06 | 38.70 | 51.66 | 36.67 | 7.50 | 22.09 |

Table 5: Model performance on the connectivity, cycle, and shortest path tasks with our proposed BAG and ALGORITHM prompting. On the two easier tasks, cycle and shortest path, our instruction-based prompting techniques resulted in a 3.07% to 16.85% performance gain across the two tasks. More complex graph reasoning tasks such as Hamilton path remain an open research question.

at reasoning on complex graph reasoning problems remains an open research question. We further explore different variants of the instructions in Appendix D. The NLGraph benchmark also empowers the evaluation of future instructions and solutions towards making LLMs better graph reasoners.

## 6   Related Work

**LLMs for tasks with implicit graphical structures.**   Previous works explored the use of LLMs on tasks with implicit graph structures: Huang et al. [2022] find that LLMs have the ability to ground high-level tasks (e.g. "make breakfast") to a set of actionable steps (e.g. "open fridge") in structured synthetic environments. Valmeekam et al. [2022] explore the possibility of using LLMs for commonsense planning. In theory-of-mind reasoning, Adhikari et al. [2020], Ammanabrolu and Riedl [2021], Sclar et al. [2023] maintain and update structured world representations as the world state change to operate in interactive and situated environments. In structured commonsense reasoning, where LLMs are given a natural language input and then asked to generate a graph such as an event graph [Tandon et al., 2019], a reasoning-graph [Madaan et al., 2021] or an argument explanation graph [Saha et al., 2021], Madaan et al. [2022] find that a code language model (CODEX) with tasks framed as code generation tasks outperforms other strong LMs. In multi-hop question answering or knowledge probing, LLMs implicitly find connections and paths among a vast network of entities and concepts [Creswell et al., 2023, Yu et al., 2022, Zhang et al., 2022a, He et al., 2021]. Recently, Chen et al. [2023] explore the potential of LLMs on the graph node classification task. Together these works demonstrate that LLMs are increasingly adopted for tasks and settings with implicit graphs and structures, while whether LLMs are robust at graph reasoning remains underexplored and may hinder the progress of these structure-aware applications. In this work, we propose the NLGraph benchmark as a comprehensive testbed towards evaluating the graph reasoning abilities of large language models.

**LLMs for few-shot reasoning.**   There is a long line of work on evaluating LLMs' reasoning ability in an in-context learning setting, including arithmetic reasoning, logical reasoning, common sense reasoning, and more. Particularly, simple math problem datasets such as AQUA Ling et al. [2017], GSM8K [Cobbe et al., 2021], and SVAMP [Patel et al., 2021] are used for evaluating arithmetic reasoning [He-Yueya et al., 2023, Touvron et al., 2023, Shi et al., 2023]. Welleck et al. [2021] developed NaturalProofs, a multi-domain dataset for studying mathematical reasoning in natural language, while Welleck et al. [2022] study LLMs' ability to generate the next step in mathematical proof and generate full proofs. LLMs have also been evaluated on logical reasoning tasks, including symbolic tasks like Coin Flip and Last Letter Concatenation [Wei et al., 2022] and Logic Grid Puzzle on the BIG-BENCH [Srivastava et al., 2023]. Commonsense reasoning datasets [Talmor et al., 2019, Geva et al., 2021] are also proposed to evaluate large language models. Most relevant to our work, various proposals to evaluate and augment the algorithm reasoning abilities of large language models are explored [Zhou et al., 2022, Lee and Kim, 2023, Zelikman et al., 2023, Liu et al., 2023]. In this work, we focus on evaluating and enhancing LLMs on graph-based reasoning and algorithmic tasks inspired by the increasing usage of LLMs in tasks with implicit graphs and structures.

## 7   Conclusion

In this work, we investigate whether LLMs are capable of explicit graph reasoning, *i.e.*, solving graph algorithm problems in natural language, across various problem categories and prompting techniques. To this end, we curate the NLGraph benchmark, a comprehensive test bed of graph-based reasoning in natural language, with 29,370 problems covering eight tasks with varying complexity. By evaluating

LLMs and prompting approaches on the NLGraph benchmark, we find that 1) LLMs do possess preliminary graph reasoning abilities, 2) the benefit of advanced prompting and in-context learning may diminish on complex reasoning tasks, while 3) LLMs are brittle to spurious correlations in problem settings. We then propose Build-a-Graph and Algorithmic Prompting, two simple instruction-based approaches that bring notable performance gains across multiple tasks. Improving LLMs' graph reasoning abilities on complex and nuanced graph reasoning tasks remains an open research question, and we encourage future work to develop upon our proposed NLGraph benchmark.

## Acknowledgements

We would like to thank the reviewers, the area chair, Jiacheng Liu, Minnan Luo, and members of the UW NLP Group for their comments and feedback. This material is funded by the DARPA Grant under Contract No. HR001120C0124. We also gratefully acknowledge support from NSF CAREER Grant No. IIS2142739, the Alfred P. Sloan Foundation Fellowship, and NSF grant No. IIS2203097. Any opinions, findings and conclusions or recommendations expressed in this material are those of the authors and do not necessarily state or reflect those of the United States Government or any agency thereof.

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

# A Discussion

**NLP tasks and graph tasks alignment.** Originally designed for processing language, LLMs are increasingly adopted for NLP tasks with structural implications. For instance, structured common-sense reasoning is similar to topological sort in that they both need to find a solution satisfying diversified constraints between entities. In the belief state graph in theory-of-mind [Adhikari et al., 2020, Ammanabrolu and Riedl, 2021], the final task is to determine the connectivity of nodes, which corresponds to the connectivity task. In multi-hop question answering [Creswell et al., 2023], LLMs implicitly find connections and paths among a vast network of entities and concepts, which resembles the shortest path task and connectivity task. However, we acknowledge that not all graph reasoning tasks in the NLGraph benchmark are clearly aligned with certain applications, thus advanced graph reasoning tasks in the NLGraph benchmark could also be viewed as a test of language model reasoning ability.

**The NLGraph benchmark envisioned as reasoning benchmark.** In addition to evaluating graph reasoning abilities, the NLGraph benchmark can also be viewed as a structural benchmark to evaluate language model reasoning. While ample LM reasoning datasets exist [Cobbe et al., 2021, Patel et al., 2021, Talmor et al., 2019], it is difficult to avoid train-test overlap since problems such as grade school math [Cobbe et al., 2021] and commonsense reasoning [Talmor et al., 2019] might be readily available in pretraining corpora, making it less convincing for evaluating reasoning abilities. On the contrary, the synthetic graph reasoning problems and their answers are highly unlikely to have exact matches in the training corpora, which makes NLGraph a more robust benchmark towards evaluating language model reasoning.

**Fine-tuning to elicit graph reasoning abilities.** To further study the graphical thinking abilities of LLMs, we envision fine-tuning as a possible direction for future work. We plan to fine-tune language models on the chain-of-thought reasoning process towards graph-based tasks on both single- and multi-task settings to see if fine-tuning might lead to enhanced graph reasoning abilities.

# B Limitations

**Tasks in NLGraph benchmark are not complete.** While we have incorporated eight graph reasoning tasks with varying complexity in the NLGraph benchmark, there are many more important graph algorithms that test different levels of graph reasoning abilities. It might also be interesting to see how LLMs perform on other graph tasks such as finding the Eulerian path, the minimum spanning tree, and the cut-edge.

**Language models that we evaluate are not complete.** We only consider four black-box LLMs, TEXT-DAVINCI-003, CODE-DAVINCI-002, GPT-3.5-TURBO, and GPT-4 in our experiments. Since we will make the NLGraph benchmark publicly available, we leave it to future work on evaluating the graph reasoning abilities of other open-source LLMs.

**Limited dataset size.** Due to monetary costs, we only evaluate LLMs on the standard version of the NLGraph benchmark which has 5,902 problems throughout the paper. We believe evaluating LLMs on the extended version of the NLGraph benchmark, with 5x more problems, may bring stronger proof to the findings we present.

**Methods for improving graph reasoning abilities.** The two prompting methods we present are simple instruction-based approaches, which work on easy graph reasoning problems to varying extents. However, on more complex graph reasoning problems, as the overall graph reasoning abilities of LLMs are limited, the instruction-based methods seem to have only marginal effects. In future work, we hope to investigate methods such as asking LLMs to generate and execute code, or simulating steps of algorithmic solutions while maintaining the state variables.

# C NLGraph Details

In Table 6, we provide the details of the NLGraph benchmark including edge probabilities and specific values of parameters mentioned in §2.2.

| Subset | Connect. | Cycle | Topo. Sort | Shortest Path | Maximum Flow | Bipartite Graph | Hamilton Path | GNNs |
|---|---|---|---|---|---|---|---|---|
| # Easy | $p$: 0.3, 0.7, 1.0 | $m$: 1-4 | $p$: 0.3, 0.5, 0.7 | $p$: 0.5, 0.7, 0.9; $w$: 1-4; $\ell$: 2-6 | $p$: 0.2, 0.3; $c$: 1-10 | $p$: 0.3-0.7 | $p$: 0.4, 0.6 | $p$: 0.4; $\ell$: 1 |
| # Medium | $p$: 0.3, 0.7, 1.0 | $m$: 1-4 | $p$: 0.3, 0.5, 0.7 | / | / | / | / | / |
| # Hard | $p$: 0.3, 0.7 | $m$: 1-4 | $p$: 0.3, 0.5 | $p$: 0.2, 0.25; $w$: 1-10; $\ell$: 2-6 | $p$: 0.25; $c$: 1-20 | $p$: 0.2-0.6 | $p$: 0.4, 0.6 | $p$: 0.2; $\ell$: 1 |

Table 6: Details of the NLGraph benchmark. $p$ denotes the edge probability. Other characters have the same meaning mentioned in §2.2.

| Method | Shortest Path | | | |
|---|---|---|---|---|
| | Easy | Hard | Easy (PC) | Hard (PC) |
| BAG | **67.78** | **33.50** | 79.20 | **42.56** |
| BAG-DOT | 65.00 | 29.00 | 75.55 | 37.18 |
| ALGORITHMIC-DOT | 66.67 | 31.00 | 78.27 | 39.68 |
| INSTRUCTION #1 | 66.22 | 32.50 | 78.18 | 40.32 |
| INSTRUCTION #2 | **67.78** | 28.50 | 77.32 | 37.06 |
| INSTRUCTION #3 | **67.78** | 30.50 | **79.56** | 37.89 |

Table 7: The results of variants of instructions on the shortest path task. The BAG prompting approach achieves the best performance across three of the four settings.

# D   Analysis

**Instruction variants**  To further study the effect of BAG prompting and ALGORITHMIC prompting, we replace the instructions with only a sequence of dots (...) equal to the number of characters in the instruction. This is to investigate whether the improved performance is attributed to the natural language instructions or simply increased computation. We also replace the BAG prompting with three other instructions not directly related to graph problems, specifically, *Let's think step by step* (INSTRUCTION 1), *Examine each detail carefully* (INSTRUCTION 2), *Think it through systematically* (INSTRUCTION 3). As illustrated in Table 7, the BAG prompting method generally outperforms the instruction variants, indicating the effectiveness of this method.

**Graph definition variants**  We study how replacing the abstract descriptions of graphs with real-world objects would impact model performance. Specifically, for the shortest path task, we change all the "node"s into "city"s, "edge"s into "road"s, and "weight"s into "distance"s. As shown in Table 9, the model performance improves on the easy subset but drops on the hard subset. While language models are indeed sensitive to the specific instantiations of the graph description, the results are mixed as to which is easier or harder.

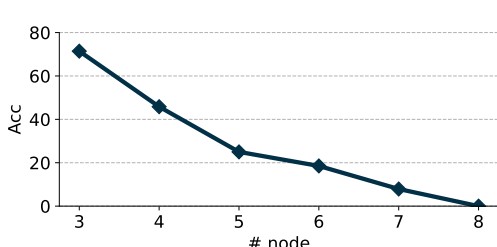

Figure 6: Model performance for the shortest path task with different numbers of nodes on the shortest path. The performance steadily drops when the path length increases.

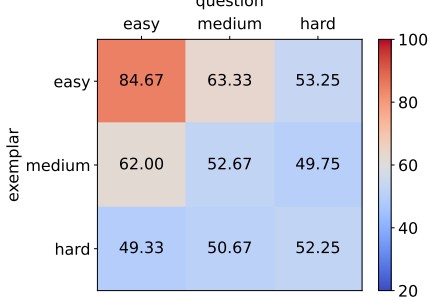

Figure 7: Model performance for the cycle task with varied exemplar difficulty. When the in-context exemplars are based on more difficult problems, the performance becomes worse.

| Method | Cycle | | | Shortest Path | | | | Hamilton Path | |
|---|---|---|---|---|---|---|---|---|---|
| | Easy | Medium | Hard | Easy | Hard | Easy (PC) | Hard (PC) | Easy | Hard |
| RANDOM | 50.00 | 50.00 | 50.00 | 50.00 | 50.00 | 6.07 | 6.69 | 0.00 | 0.00 |
| FEW-SHOT | 58.00 | 50.00 | 49.75 | 39.44 | 28.50 | 55.10 | 37.63 | 36.67 | 7.00 |
| CoT | 60.67 | 66.83 | 69.00 | 41.11 | / | 45.00 | / | 50.00 | 12.00 |
| CoT+SC | 59.33 | 52.67 | 53.75 | 42.22 | / | 49.26 | / | 34.00 | 5.00 |

Table 8: The results of CODE-DAVINCI-002 on the cycle, shortest path, and Hamilton path task.

| Method | Shortest Path | | | |
|---|---|---|---|---|
| | Easy | Hard | Easy (PC) | Hard (PC) |
| RANDOM | 50.00 | 50.00 | 6.07 | 6.69 |
| ZERO-SHOT | 29.40 | 14.00 | 47.53 | 20.13 |
| FEW-SHOT | 41.11 | 20.50 | 58.78 | 29.28 |
| CoT | 73.33 | 28.00 | 82.57 | 35.50 |
| 0-CoT | 6.67 | 2.00 | 65.40 | 51.95 |
| CoT+SC | 72.78 | 27.00 | 83.50 | 35.19 |

Table 9: Model performance when graph descriptions are changed into cities, roads, and distances on the shortest path task.

**LLMs are brittle to problem scales**   For the shortest path task, we study the correlation between model performance and the number of nodes on the optimal shortest path. We group problems by the shortest path length and present results in Figure 6 for the CoT approach, which shows that the accuracy steadily drops when the number of nodes on the shortest path increases. While the reasoning process towards finding the shortest paths stays the same for optimal solutions with varying lengths, large language models are not robust to changes in graph scales and problem settings.

**In-context exemplars and test difficulty**   We investigate the model performance when the in-context exemplars and test difficulty vary. For the cycle task, we write chain-of-thought solutions for problems selected from the medium and hard subsets, and evaluate TEXT-DAVINCI-003 on the cycle task with the two prompts. We present the results in Figure 7. The performance drops when the exemplar difficulty increases, indicating that LLM struggles to learn from exemplars with more difficulty.

**Partial credit for the maximum flow task**
We provide partial credit for the maximum flow task in Table 10. The performance generally has the same trend as the results provided in Figure 2. Though on the hard subsets, the performance of CoT is the highest, we believe this is more like guessing, as when CoT is combined with self-consistency, the performance drops significantly.

| Method | Easy | Hard |
|---|---|---|
| ZERO-SHOT | **25.65** | 10.37 |
| FEW-SHOT | 24.83 | 12.37 |
| CoT | 16.99 | **22.07** |
| 0-CoT | 17.41 | 7.04 |
| CoT+SC | 9.88 | 8.62 |

Table 10: Partial credit for the maximum flow task.

# E   Additional Large Language Models

## E.1   CODE-DAVINCI-002 Results

We evaluate CODE-DAVINCI-002 on cycle, shortest path, and Hamilton path. The results are provided in Table 8. The findings are mostly similar to TEXT-DAVINCI-003: language models are preliminary graph reasoners on easy tasks while the benefit of in-context learning and advanced prompting is limited on more complex problems. However, in the CoT and CoT+SC settings, CODE-DAVINCI-002 generates too many tokens before reaching the final answer, so we do not present the results in Table 8.

## E.2 OPT Results

We fine-tune OPT-2.7B [Zhang et al., 2022b] and evaluate the model on the NLGraph benchmark. We randomly divide the standard version of the NLGraph benchmark into 4902 training problems and 1,000 test problems. We generate zero-shot solutions for the training problems, combine the solutions with the problems, and attain nine corpus of text (*i.e.* one for each task and one that concatenating all the text). We then fine-tune OPT on the nine corpus separately. However, we find that fine-tuning did not elicit the graph reasoning abilities of OPT-2.7B: the generated text is mostly meaningless, falling short of answering questions and following formats. We leave it to future work on fine-tuning larger language models to elicit graph reasoning abilities due to our limitations in computational resources.

## E.3 Qualitative Analaysis with GPT-3.5-TURBO and GPT-4

Due to monetary costs, we only conduct qualitative analysis on GPT-3.5-TURBO and GPT-4 with 19 representative problems across the eight tasks. We provide the results in Table 11, Table 12, Table 13, Table 14, Table 15. Results demonstrate that while GPT-4 does present better graph reasoning abilities, it is still far from being a robust reasoner towards graph-based problems.

Table 11: Results of GPT-3.5-TURBO and GPT-4 on 19 cases (#1 to #6 example) through the eight tasks. We omitted the CoT in-context exemplars in the table.

| Question | GPT-3.5-TURBO | GPT-4 |
|---|---|---|
| Determine if there is a path between two nodes in the graph. Note that (i,j) means that node i and node j are connected with an undirected edge. Graph: (0,4) (1,3) (1,2) (1,6) (2,5) (2,6) (3,5) (5,6) 
 Q: Is there a path between node 4 and node 5? | Node 4 is not directly connected to node 5, but we can follow the path: 4->0->6->2->5, so the answer is yes. ✗ | Node 4 is only connected to node 0, which does not have any connections to the other nodes in the graph. Node 5 is in the connected block consisting of nodes 1, 2, 3, 5, and 6. Node 4 and node 5 are not in the same connected block, so the answer is no. ✔ |
| Determine if there is a path between two nodes in the graph. Note that (i,j) means that node i and node j are connected with an undirected edge. Graph: (0,3) (0,5) (0,11) (0,8) (0,6) (0,1) (1,10) (1,3) (1,5) (1,9) (1,6) (1,2) (2,10) (2,3) (2,5) (2,9) (2,8) (2,7) (2,6) (2,13) (3,9) (3,11) (3,8) (3,7) (4,12) (5,10) (5,9) (5,8) (5,7) (5,6) (6,10) (6,9) (6,11) (6,8) (7,10) (7,11) (7,8) (7,13) (8,10) (8,11) (8,13) (9,10) (9,11) (9,13) (10,11) (10,13) 
 Q: Is there a path between node 6 and node 3? | Node 6 is connected to node 0, node 0 is connected to node 3. We can follow the path: 6->0->3, so the answer is yes. ✔ | A: Node 6 is connected to node 2, node 2 is connected to node 3. We can follow the path: 6->2->3, so the answer is yes. ✔ |
| In an undirected graph, (i,j) means that node i and node j are connected with an undirected edge. The nodes are numbered from 0 to 7, and the edges are: (2,4) (0,3) (1,2) (5,7) (1,5) (6,5) (1,0) 
 Q: Is there a cycle in this graph? | The edges (1,2), (2,4), (4,0), (0,3), (3,1) form a cycle, so yes, there is a cycle in this graph. ✗ | The edges (1,0), (1,2), (1,5), (6,5), (5,7) do not form a cycle. However, the edges (1,0), (1,2), (2,4) do form a cycle. So yes, there is a cycle in this graph. ✗ |
| In an undirected graph, (i,j) means that node i and node j are connected with an undirected edge. The nodes are numbered from 0 to 7, and the edges are: (1,7) (0,6) (1,3) (4,5) (1,0) (0,4) (0,2) (7,2) 
 Q: Is there a cycle in this graph? | The edges (0,6), (6,4), (4,5), (5,7), (7,1), (1,0) form a cycle, so yes, there is a cycle in this graph. ✗ | The edges (1,7), (7,2), (2,0), (0,1) form a cycle, so yes, there is a cycle in this graph. ✔ |
| In an undirected graph, the nodes are numbered from 0 to 5, and the edges are: An edge between node 2 and node 0 with weight 2, An edge between node 0 and node 4 with weight 1, An edge between node 1 and node 2 with weight 4, An edge between node 1 and node 3 with weight 4, An edge between node 4 and node 1 with weight 4, An edge between node 2 and node 3 with weight 1, An edge between node 2 and node 5 with weight 2, An edge between node 2 and node 4 with weight 3, An edge between node 5 and node 3 with weight 2, An edge between node 3 and node 4 with weight 3, An edge between node 4 and node 5 with weight 2. 
 Q: Give the shortest path from node 3 to node 0. | All the paths from node 3 to node 0 are: 3,2,0 with a total weight of 2 + 2 = 4, 3,2,4,1,0 with a total weight of 3 + 4 + 1 = 8, 3,2,4,0 with a total weight of 3 + 1 = 4, 3,1,4,0 with a total weight of 4 + 4 + 1 = 9, 3,1,2,0 with a total weight of 4 + 4 + 2 = 10, 3,5,2,0 with a total weight of 2 + 2 + 2 = 6, 3,5,4,0 with a total weight of 2 + 2 + 1 = 5. The weight of path 3,2,0 and 3,5,4,0 is the smallest, so the shortest path from node 3 to node 0 is either 3,2,0 or 3,5,4,0, both with a total weight of 4. ✗ | All the paths from node 3 to node 0 are: 3,2,0 with a total weight of 1 + 2 = 3, 3,1,2,0 with a total weight of 4 + 4 + 2 = 10, 3,4,0 with a total weight of 3 + 1 = 4, 3,4,2,0 with a total weight of 3 + 3 + 2 = 8, 3,5,2,0 with a total weight of 2 + 2 + 2 = 6, 3,5,4,0 with a total weight of 2 + 2 + 1 = 5. The weight of path 3,2,0 is the smallest, so the shortest path from node 3 to node 0 is 3,2,0 with a total weight of 3. ✔ |
| In an undirected graph, the nodes are numbered from 0 to 4, and the edges are: An edge between node 1 and node 0 with weight 1, An edge between node 2 and node 0 with weight 1, An edge between node 4 and node 1 with weight 1, An edge between node 1 and node 2 with weight 3, An edge between node 2 and node 3 with weight 3. 
 Q: Give the shortest path from node 3 to node 1. | All the paths from node 3 to node 1 are: 3,2,1 with a total weight of 3 + 3 = 6, 3,2,0,1 with a total weight of 1 + 1 + 1 = 3, 3,2,1,0,1 with a total weight of 3 + 1 + 1 + 1 = 6, 3,2,0,1,4,1 with a total weight of 1 + 1 + 1 + 1 + 1 = 5. The weight of path 3,2,0,1 is the smallest, so the shortest path from node 3 to node 1 is 3,2,0,1 with a total weight of 3. ✗ | All the paths from node 3 to node 1 are: 3,2,1 with a total weight of 3 + 3 = 6, 3,2,0,1 with a total weight of 3 + 1 + 1 = 5. The weight of path 3,2,0,1 is the smallest, so the shortest path from node 3 to node 1 is 3,2,0,1 with a total weight of 5. ✔ |

Table 12: Results of GPT-3.5-TURBO and GPT-4 on 19 cases (#7 to #9 example ) through the eight tasks. We omitted the CoT in-context exemplars in the table.

| Question | GPT-3.5-TURBO | GPT-4 |
|---|---|---|
| In an undirected graph, the nodes are numbered from 0 to 6, and the edges are: An edge between node 0 and node 2 with weight 1, An edge between node 3 and node 0 with weight 4, An edge between node 5 and node 0 with weight 2, An edge between node 4 and node 0 with weight 3, An edge between node 0 and node 1 with weight 4, An edge between node 0 and node 6 with weight 2, An edge between node 1 and node 2 with weight 2, An edge between node 3 and node 1 with weight 3, An edge between node 4 and node 1 with weight 3, An edge between node 1 and node 6 with weight 2, An edge between node 2 and node 5 with weight 4, An edge between node 2 and node 4 with weight 3, An edge between node 6 and node 2 with weight 2, An edge between node 3 and node 5 with weight 1, An edge between node 4 and node 3 with weight 2, An edge between node 6 and node 4 with weight 4, An edge between node 4 and node 5 with weight 3, An edge between node 6 and node 5 with weight 2. Q: Give the shortest path from node 2 to node 3. | All the paths from node 2 to node 3 are: 2,0,3 with a total weight of 1 + 4 = 5, 2,1,3 with a total weight of 2 + 3 = 5, 2,4,3 with a total weight of 3 + 2 = 5. The weight of path 2,0,3 is the smallest, so the shortest path from node 2 to node 3 is 2,0,3 with a total weight of 5. ✗ | All the paths from node 2 to node 3 are: 2,0,3 with a total weight of 1 + 4 = 5, 2,1,3 with a total weight of 2 + 3 = 5, 2,1,0,3 with a total weight of 2 + 4 + 4 = 10, 2,1,6,3 with a total weight of 2 + 2 + 4 = 8, 2,4,3 with a total weight of 3 + 2 = 5, 2,4,0,3 with a total weight of 3 + 3 + 4 = 10, 2,4,1,3 with a total weight of 3 + 3 + 3 = 9, 2,4,1,0,3 with a total weight of 3 + 3 + 4 + 4 = 14, 2,4,1,6,3 with a total weight of 3 + 3 + 2 + 4 = 12, 2,5,0,3 with a total weight of 4 + 2 + 4 = 10, 2,5,4,3 with a total weight of 4 + 5 + 2 = 11, 2,5,4,0,3 with a total weight of 4 + 5 + 3 + 4 = 16, 2,5,4,1,3 with a total weight of 4 + 5 + 3 + 3 = 15, 2,5,6,3 with a total weight of 4 + 6 + 4 = 14, 2,6,1,3 with a total weight of 2 + 2 + 3 = 7, 2,6,1,0,3 with a total weight of 2 + 2 + 4 + 4 = 12, 2,6,5,0,3 with a total weight of 2 + 5 + 2 + 4 = 13, 2,6,5,4,3 with a total weight of 2 + 5 + 5 + 2 = 14. The weight of paths 2,0,3, 2,1,3, and 2,4,3 is the smallest. So, there are three shortest paths from node 2 to node 3, which are 2,0,3; 2,1,3; and 2,4,3, each with a total weight of 5. ✗ |
| In a directed graph with 5 nodes numbered from 0 to 4: node 2 should be visited before node 4 node 2 should be visited before node 1 node 2 should be visited before node 0 node 4 should be visited before node 1 node 4 should be visited before node 0 Q: Can all the nodes be visited? Give the solution. | This is not possible as there is a cycle between nodes 0, 1, and 4. Node 2 cannot be visited before nodes 0, 1, and 4, and nodes 0, 1, and 4 cannot be visited before node 2. Therefore, it is not possible to visit all the nodes in this graph. ✗ | Since no nodes are required to be visited before node 2 and node 3, node 2 and node 3 can be visited first. Since visiting node 4 requires visiting node 2 first, after node 2 is visited, node 4 can be visited. Since visiting node 0 requires visiting node 2, node 4 first, after node 2 and node 4 are visited, node 0 can be visited. Since visiting node 1 requires visiting node 2, node 4 first, after node 2 and node 4 are visited, node 1 can be visited. So the solution is: 2,3,4,0,1. ✔ |
| In a directed graph with 5 nodes numbered from 0 to 4: node 0 should be visited before node 1 node 1 should be visited before node 2 node 3 should be visited before node 1 node 3 should be visited before node 2 node 4 should be visited before node 3 node 4 should be visited before node 0 node 4 should be visited before node 1 Q: Can all the nodes be visited? Give the solution. | This graph has a cycle, which means it is not possible to visit all nodes in a valid order. Specifically, there is a cycle between nodes 1, 2, and 3, which cannot be resolved based on the given constraints. ✗ | Since no nodes are required to be visited before node 4, node 4 can be visited first. Since visiting node 0 and node 3 requires visiting node 4 first, after node 4 is visited, node 0 and node 3 can be visited. Since visiting node 1 requires visiting node 0 and node 3 first, after node 0 and node 3 are visited, node 1 can be visited. Since visiting node 2 requires visiting node 1 and node 3 first, after node 1 and node 3 are visited, node 2 can be visited. So the solution is: 4,0,3,1,2. ✔ |

Table 13: Results of GPT-3.5-TURBO and GPT-4 on 19 cases (#10 to #12 example ) through the eight tasks. We omitted the COT in-context exemplars in the table.

| Question | GPT-3.5-TURBO | GPT-4 |
|---|---|---|
| In a directed graph with 6 nodes numbered from 0 to 5: node 0 should be visited before node 5 node 0 should be visited before node 1 node 2 should be visited before node 1 node 3 should be visited before node 0 node 3 should be visited before node 5 node 3 should be visited before node 1 node 4 should be visited before node 0 node 4 should be visited before node 2 node 5 should be visited before node 2 Q: Can all the nodes be visited? Give the solution | Since no nodes are required to be visited before node 3 and node 4, node 3 and node 4 can be visited first. Since visiting node 0 requires visiting node 3 first, after node 3 is visited, node 0 can be visited. Since visiting node 1 requires visiting node 0, node 2, node 3 first, after node 0, node 2, and node 3 are visited, node 1 can be visited. Since visiting node 5 requires visiting node 0, node 3 first, after node 0 and node 3 are visited, node 5 can be visited. Since visiting node 2 requires visiting node 4, node 5 first, after node 4 and node 5 are visited, node 2 can be visited. So the solution is: 3,4,0,2,1,5. ✗ | Since no nodes are required to be visited before node 3 and node 4, node 3 and node 4 can be visited first. Since visiting node 0 requires visiting node 3 and node 4 first, after node 3 and node 4 are visited, node 0 can be visited. Since visiting node 5 requires visiting node 0 and node 3 first, after node 0 and node 3 are visited, node 5 can be visited. Since visiting node 1 requires visiting node 0, node 2, and node 3 first, node 2 must be visited before node 1. Since visiting node 2 requires visiting node 4 and node 5 first, after node 4 and node 5 are visited, node 2 can be visited. Now, since node 0, node 2, and node 3 have been visited, node 1 can be visited. So the solution is: 3,4,0,5,2,1. ✔ |
| In a directed graph, the nodes are numbered from 0 to 5, and the edges are: an edge from node 1 to node 3 with capacity 4, an edge from node 1 to node 0 with capacity 9, an edge from node 2 to node 1 with capacity 4, an edge from node 3 to node 1 with capacity 4, an edge from node 4 to node 3 with capacity 1, an edge from node 4 to node 5 with capacity 2, an edge from node 5 to node 1 with capacity 4, an edge from node 5 to node 0 with capacity 7. Q: What is the maximum flow from node 4 to node 0? | There is no path from node 4 to node 0, so the maximum flow from node 4 to node 0 is 0. ✗ | From the source (node 4), we can send 1 unit of flow to node 3, and 2 units of flow to node 5. 1 unit of flow arrives at node 3, then we can send 1 unit of flow from node 3 to node 1. 2 units of flow arrive at node 5, then we can send 2 units of flow from node 5 to node 0. 1 unit of flow arrives at node 1, then we can send 1 unit of flow from node 1 to node 0. The total unit of flow arrives at node 0 is 3. So the maximum flow from node 4 to node 0 is 3. ✔ |
| In a directed graph, the nodes are numbered from 0 to 7, and the edges are: an edge from node 0 to node 6 with capacity 1, an edge from node 0 to node 7 with capacity 10, an edge from node 2 to node 1 with capacity 9, an edge from node 2 to node 7 with capacity 8, an edge from node 3 to node 1 with capacity 9, an edge from node 3 to node 6 with capacity 10, an edge from node 3 to node 4 with capacity 7, an edge from node 4 to node 6 with capacity 1, an edge from node 5 to node 3 with capacity 7, an edge from node 5 to node 2 with capacity 9, an edge from node 6 to node 4 with capacity 1, an edge from node 7 to node 5 with capacity 8, an edge from node 7 to node 6 with capacity 5. Q: What is the maximum flow from node 2 to node 4? | From the source (node 2), we can send 8 units of flow to node 7, and 9 units of flow to node 1. 8 units of flow arrive at node 7, then we can send 5 units of flow from node 7 to node 6, and 3 units of flow from node 7 to node 5. 9 units of flow arrive at node 1, then we can send 9 units of flow from node 3 to node 1. 9 units of flow arrive at node 6, then we can send 1 unit of flow from node 6 to node 4. 7 units of flow arrive at node 3, then we can send 7 units of flow from node 3 to node 4. The total unit of flow arrives at node 4 is 8. So the maximum flow from node 2 to node 4 is 8. ✔ | From the source (node 2), we can send 9 units of flow to node 1, and 8 units of flow to node 7. 9 units of flow arrive at node 1, then we can send 9 units of flow from node 1 to node 3. 8 units of flow arrive at node 7, then we can send 8 units of flow from node 7 to node 5, and 5 units of flow from node 7 to node 6. 8 units of flow arrive at node 5, then we can send 7 units of flow from node 5 to node 3, and 1 unit of flow from node 5 to node 2. 14 units of flow arrive at node 3, then we can send 14 units of flow from node 3 to node 4. The total unit of flow arrives at node 4 is 14. So the maximum flow from node 2 to node 4 is 14. ✗ |

Table 14: Results of GPT-3.5-TURBO and GPT-4 on 19 cases (#13 to #16 example ) through the eight tasks. We omitted the COT in-context exemplars in the table.

| Question | GPT-3.5-TURBO | GPT-4 |
|---|---|---|
| In a directed graph, the nodes are numbered from 0 to 7, and the edges are: an edge from node 0 to node 6 with capacity 9, an edge from node 0 to node 3 with capacity 7, an edge from node 1 to node 7 with capacity 8, an edge from node 1 to node 2 with capacity 3, an edge from node 2 to node 7 with capacity 7, an edge from node 2 to node 4 with capacity 5, an edge from node 3 to node 5 with capacity 7, an edge from node 3 to node 1 with capacity 8, an edge from node 4 to node 0 with capacity 3, an edge from node 5 to node 7 with capacity 5, an edge from node 6 to node 5 with capacity 6, an edge from node 6 to node 7 with capacity 6, an edge from node 6 to node 0 with capacity 9, an edge from node 7 to node 4 with capacity 4, an edge from node 7 to node 1 with capacity 8. Q: What is the maximum flow from node 1 to node 6? | There is no path from node 1 to node 6 in this graph, so the maximum flow from node 1 to node 6 is 0. ✗ | From the source (node 1), we can send 8 units of flow to node 7, and 3 units of flow to node 2. 8 units of flow arrive at node 7, then we can send 4 units of flow from node 7 to node 4, and 4 units of flow from node 7 to node 1. 4 units of flow arrive at node 4, then we can send 3 units of flow from node 4 to node 0. 3 units of flow arrive at node 0, then we can send 3 units of flow from node 0 to node 6. 3 units of flow arrive at node 2, then we can send 3 units of flow from node 2 to node 4. 3 units of flow arrive at node 4, then we can send 3 units of flow from node 4 to node 0. 3 units of flow arrive at node 0, then we can send 3 units of flow from node 0 to node 6. The total unit of flow arrives at node 6 is 6. So the maximum flow from node 1 to node 6 is 6. ✗ |
| In an undirected graph, (i,j) means that node i and node j are connected with an undirected edge. The nodes are numbered from 0 to 6, and the edges are: (4,3) (6,5) (5,1) (3,0) (2,4) (1,0) (5,2) (4,0) (2,3) (0,2) (6,2) Q: Is there a path in this graph that visits every node exactly once? If yes, give the path. Note that in a path, adjacent nodes must be connected with edges. | No, there is no path that visits every node exactly once. This graph contains two disconnected components: 0,1,2,3,4 and 5,6. It is not possible to visit all nodes in a single path because there is no edge connecting the two components. ✗ | Yes. We can start at node 3. As node 3 is connected with node 0, and node 0 is not visited, we can then visit node 0. As node 0 is connected with node 1, and node 1 is not visited, we can then visit node 1. As node 1 is connected with node 5, and node 5 is not visited, we can then visit node 5. As node 5 is connected with node 6, and node 6 is not visited, we can then visit node 6. As node 6 is connected with node 2, and node 2 is not visited, we can then visit node 2. As node 2 is connected with node 4, and node 4 is not visited, we can then visit node 4. The path can be: 3, 0, 1, 5, 6, 2, 4. ✔ |
| In an undirected graph, (i,j) means that node i and node j are connected with an undirected edge. The nodes are numbered from 0 to 4, and the edges are: (1,3) (1,2) (4,3) (2,0) (0,4) (0,1) Q: Is there a path in this graph that visits every node exactly once? If yes, give the path. Note that in a path, adjacent nodes must be connected with edges. | No, there is no path in this graph that visits every node exactly once. This is because node 1 has degree 2, meaning it is connected to two other nodes. In order to visit every node exactly once, we would need to start and end at node 1, but once we leave node 1, we cannot return to it without visiting another node twice. Therefore, it is impossible to find a path that visits every node exactly once in this graph. ✗ | Yes. We can start at node 0. As node 0 is connected with node 1, and node 1 is not visited, we can then visit node 1. As node 1 is connected with node 3, and node 3 is not visited, we can then visit node 3. As node 3 is connected with node 4, and node 4 is not visited, we can then visit node 4. As node 4 is connected with node 2, and node 2 is not visited, we can then visit node 2. The path can be: 0,1,3,4,2. ✗ |
| There are 4 job applicants numbered from 0 to 3, and 2 jobs numbered from 0 to 1. Each applicant is interested in some of the jobs. Each job can only accept one applicant and a job applicant can be appointed for only one job. Applicant 0 is interested in job 1. Applicant 2 is interested in job 0. Applicant 0 is interested in job 0. Applicant 2 is interested in job 1. Applicant 3 is interested in job 1. Applicant 3 is interested in job 0. Applicant 1 is interested in job 0. Q: Find an assignment of jobs to applicants in such that the maximum number of applicants find the job they are interested in. | Applicant 0 is interested in job 0, job 1. Applicant 1 is interested in job 0. Applicant 2 is interested in job 0, job 1. Applicant 3 is interested in job 0, job 1. We can make the following assignments: applicant 0: job 1 applicant 2: job 0 applicant 3: job 1 3 applicants can find the job they are interested in. ✗ | Applicant 0 is interested in job 0, job 1. Applicant 1 is interested in job 0. Applicant 2 is interested in job 0, job 1. Applicant 3 is interested in job 0, job 1. We can make the following assignments: applicant 0: job 0 applicant 1: - (no job available) applicant 2: job 1 applicant 3: - (no job available) 2 applicants can find the job they are interested in. ✔ |

Table 15: Results of GPT-3.5-TURBO and GPT-4 on 19 cases (#17 to #19 example ) through the eight tasks. We omitted the CoT in-context exemplars in the table.

| Question | GPT-3.5-TURBO | GPT-4 |
|---|---|---|
| There are 5 job applicants numbered from 0 to 4, and 6 jobs numbered from 0 to 5. Each applicant is interested in some of the jobs. Each job can only accept one applicant and a job applicant can be appointed for only one job.
Applicant 0 is interested in job 3.
Applicant 4 is interested in job 5. Applicant 4 is interested in job 4.
Applicant 3 is interested in job 0.
Applicant 0 is interested in job 5.
Applicant 1 is interested in job 0.
Applicant 0 is interested in job 2.
Applicant 0 is interested in job 4.
Applicant 4 is interested in job 0.
Applicant 4 is interested in job 2.
Applicant 2 is interested in job 3.
Q: Find an assignment of jobs to applicants in such that the maximum number of applicants find the job they are interested in. | Applicant 0 is interested in job 2, job 3, job 4, job 5. Applicant 1 is interested in job 0. Applicant 2 is interested in job 3. Applicant 3 is interested in job 0. Applicant 4 is interested in job 0, job 2, job 4, job 5. We can make the following assignments: applicant 0: job 5 applicant 1: job 0 applicant 2: job 3 applicant 3: job 0 applicant 4: job 2 5 applicants can find the job they are interested in. ✗ | Applicant 0 is interested in job 2, job 3, job 4, job 5. Applicant 1 is interested in job 0. Applicant 2 is interested in job 3. Applicant 3 is interested in job 0. Applicant 4 is interested in job 0, job 2, job 4, job 5. We can make the following assignments: applicant 0: job 3 applicant 1: job 0 applicant 4: job 4 3 applicants can find the job they are interested in. ✗ |
| n an undirected graph, the nodes are numbered from 0 to 5, and every node has an embedding. (i,j) means that node i and node j are connected with an undirected edge.
Embeddings:
node 0: [1,1]
node 1: [0,1]
node 2: [1,0]
node 3: [1,1]
node 4: [0,0]
node 5: [0,1]
The edges are: (0,1) (5,0) (1,4) (1,5) (2,4) (4,3)
In a simple graph convolution layer, each node's embedding is updated by the sum of its neighbors' embeddings.
Q: What's the embedding of each node after one layer of simple graph convolution layer? | Layer 1: node 0: sum of embeddings of node 1, node 5, [0,1] + [0,1] = [0,2].
node 1: sum of embeddings of node 0, node 4, node 5, [1,1] + [0,0] + [0,1] = [1,2].
node 2: sum of embeddings of node 4, [0,0] = [0,0].
node 3: sum of embeddings of node 4, [0,0] = [0,0].
node 4: sum of embeddings of node 1, node 2, node 3, [1,2] + [1,0] + [0,0] = [2,2].
node 5: sum of embeddings of node 0, node 1, [0,2] + [1,2] = [1,4].
So the answer is:
node 0: [0,2]
node 1: [1,2]
node 2: [0,0]
node 3: [0,0]
node 4: [2,2]
node 5: [1,4] ✗ | Layer 1:
node 0: sum of embeddings of node 1, node 5, [0,1] + [0,1] = [0,2].
node 1: sum of embeddings of node 0, node 4, node 5, [1,1] + [0,0] + [0,1] = [1,2].
node 2: sum of embeddings of node 4, [0,0] = [0,0].
node 3: sum of embeddings of node 4, [0,0] = [0,0].
node 4: sum of embeddings of node 1, node 2, node 3, [0,1] + [1,0] + [1,1] = [2,2].
node 5: sum of embeddings of node 0, node 1, [1,1] + [0,1] = [1,2].
So the answer is:
node 0: [0,2]
node 1: [1,2]
node 2: [0,0]
node 3: [0,0]
node 4: [2,2]
node 5: [1,2] ✔ |
| In an undirected graph, the nodes are numbered from 0 to 7, and every node has an embedding. (i,j) means that node i and node j are connected with an undirected edge.
Embeddings:
node 0: [0,1]
node 1: [1,1]
node 2: [1,0]
node 3: [1,1]
node 4: [1,1]
node 5: [0,0]
node 6: [1,0]
node 7: [1,0]
The edges are: (0,3) (4,0) (6,0) (5,1) (1,4) (5,2) (2,3) (2,6) (3,5) (3,6) (5,4) (7,5)
In a simple graph convolution layer, each node's embedding is updated by the sum of its neighbors' embeddings.
Q: What's the embedding of each node after one layer of simple graph convolution layer? | Layer 1: node 0: sum of embeddings of node 3, node 4, node 6, [1,1] + [1,1] + [1,0] = [3,2].
node 1: sum of embeddings of node 4, [1,1] = [1,1].
node 2: sum of embeddings of node 3, node 5, node 6, [1,1] + [0,0] + [1,0] = [2,1].
node 3: sum of embeddings of node 0, node 2, node 5, node 6, [0,1] + [1,1] + [0,0] + [1,0] = [2,2].
node 4: sum of embeddings of node 0, node 1, node 5, [3,2] + [1,1] + [0,0] = [4,3].
node 5: sum of embeddings of node 2, node 3, node 4, node 7, [2,1] + [2,2] + [4,3] + [1,0] = [9,6].
node 6: sum of embeddings of node 0, node 2, node 3, [3,2] + [1,1] + [1,0] = [5,3].
node 7: sum of embeddings of node 5, [9,6] = [9,6].
So the answer is:
node 0: [3,2]
node 1: [1,1]
node 2: [2,1]
node 3: [2,2]
node 4: [4,3]
node 5: [9,6]
node 6: [5,3]
node 7: [9,6] ✗ | Layer 1: node 0: sum of embeddings of node 3, node 4, node 6, [1,1] + [1,1] + [1,0] = [3,2].
node 1: sum of embeddings of node 5, node 4, [0,0] + [1,1] = [1,1].
node 2: sum of embeddings of node 5, node 3, node 6, [0,0] + [1,1] + [1,0] = [2,1].
node 3: sum of embeddings of node 0, node 2, node 5, node 6, [0,1] + [1,0] + [0,0] + [1,0] = [2,1].
node 4: sum of embeddings of node 0, node 1, node 5, [0,1] + [1,1] + [0,0] = [1,2].
node 5: sum of embeddings of node 1, node 2, node 3, node 4, node 7, [1,1] + [1,0] + [1,1] + [1,1] + [1,0] = [5,3].
node 6: sum of embeddings of node 0, node 2, node 3, node 7, [0,1] + [1,0] + [1,1] + [1,0] = [3,2].
node 7: sum of embeddings of node 5, [0,0] = [0,0].
So the answer is:
node 0: [3,2]
node 1: [1,1]
node 2: [2,1]
node 3: [2,1]
node 4: [1,2]
node 5: [5,3]
node 6: [3,2]
node 7: [0,0] ✗ |

