# OpenReview forum: "Can Language Models Solve Graph Problems in Natural Language?"
_NeurIPS.cc/2023/Conference — NeurIPS 2023 spotlight_

### Official Review · Reviewer_EhYM · 2023-06-19

**Soundness:** 4 excellent
**Presentation:** 4 excellent
**Contribution:** 4 excellent
**Rating:** 8
**Confidence:** 3

**Summary:**

The applications of Large Language Models (LLMs) have been extended beyond natural language, covering more complex tasks that might have an implicit graph structure, such as planning for robots. This work introduces a benchmark named Natural Language Graph (NLGraph) to examine the explicit graph reasoning of advanced LLMs.

NLGraph contains eight basic graph reasoning task, varying from easy (connectivity, cycle, topological, ...) to hard (maximum flow, Hamilton path, ...). Based on NLGraph, this work has several findings:

* Current LLM (text-davinci-003) shows preliminary graph reasoning abilities. It achieves much better results than random guess in the easy task, for example, connectivity and cycle.
* Advanced prompting fails in more complex tasks.
* LLM are not robust to spurious correlation.

They also propose two simple prompts to enhance the LLM in easy graph reasoning task.

**Strengths:**

* This paper is well-motivated and finely written. As large language models (LLMs) grow increasingly proficient, we are in need of novel, challenging benchmarks to thoroughly assess their capabilities. The graph structure exemplifies data structures that are prevalent in real life.
* This benchmark is fairly comprehensive, encompassing the eight fundamental types of graph problems. It also forms subsets of varying difficulty levels by manipulating factors such as the number of nodes and density.
* The findings are both intriguing and illuminating. For instance, the revelation that current advanced prompt strategies cannot generalize to complex tasks could potentially motivate research into exploring innovative techniques.

**Weaknesses:**

The evaluations are only conducted on text-davinci-003, therefore, some conclusions may be limited to this model.

**Questions:**

What is the correspondence between, train, main, and test splits provided in the supplementary material and the standard and extended set in the text?

**Limitations:**

The authors discuss limitations comprehensively in the supplementary material, partly as future work.

---

> ### Author Rebuttal · Authors · 2023-08-08
>
> Thank you for your time and positive feedback!
>
> > The evaluations are only conducted on text-davinci-003, therefore, some conclusions may be limited to this model.
>
> While we evaluate text-davinci-003 on the whole dataset, we also provide the results of code-davinci-002 on several tasks in the NLGraph benchmark, which show similar trends. Moreover, we give qualitative analysis results of GPT-3.5-turbo and GPT-4 in the appendix. As we will make the NLGraph benchmark publicly available, we hope future work will further explore other LMs on the NLGraph benchmark.
>
> > What is the correspondence between, train, main, and test splits provided in the supplementary material and the standard and extended set in the text?
>
> Main splits are split into train splits and test splits. The supplementary material only contains the standard set in the text, which we use to run all the experiments. We will make the extended set publicly available after the anonymity period.

---

### Official Review · Reviewer_U4cj · 2023-07-06

**Soundness:** 3 good
**Presentation:** 3 good
**Contribution:** 3 good
**Rating:** 6
**Confidence:** 4

**Summary:**

The paper curates a benchmark dataset called NLGraph that contains 8 types of graph reasoning problems. Given that various previous works have used LLMs to solve real-world tasks that implicitly require some form of simple graph reasoning, this dataset aims to isolate and analyze the ability of LLMs to graph reasoning problems. The paper tests various existing prompting strategies and also proposes two new strategies specifically useful for solving graph reasoning problems. The newly proposed strategies are evaluated on the proposed NLGraph dataset and are shown to improve the reasoning performance.

**Strengths:**

### originality
The paper is the first to curate a dataset aimed at isolating the graph reasoning abilities of LLMs in the abstract. This is complementary to all the works that use LLMs to solve tasks that implicitly require graph reasoning, and thus can provide valuable insights that can be useful in real applications.
### quality
1. The discussion about the limitations of the work (in the appendix) clearly states the scope of applicability of their finding.
2. The experiments are well-designed, especially the ones demonstrating the brittleness of LLMs.
### clarity:
The paper is well-organized, clearly written, and easy to follow.
### significance
The curated dataset and the experimental setup proposed in the paper can be a good starting point for further exploration of the graph reasoning abilities of LLMs, making this work useful for the community.

**Weaknesses:**

### quality:
1. The performance of the model depends heavily on how the prompts are constructed. While Figure 1 shows the instructions used in the prompts clearly, to ensure reproducibility, the complete prompts for more complex strategies like COT should be provided.
### Significance:
1. The motivation behind choosing the eight specific graph reasoning tasks is lacking. The ultimate goal of performing graph reasoning using LLMs is to use it to solve simple real-world tasks where graph reasoning is required. Given this motivation, the authors should provide additional justification for picking the reasoning tasks used in the paper.
2. For the findings to apply to real-world data, the proposed prompting strategies should be tested with graphs that have real-world semantics associated with notes and edges. For example, representing nodes as numbers might allow the model to easily decouple the inherent meaning of the nodes from the meaning in the context of the given graph. This might not be an easy task if the nodes were a set of words, phrases, or actions.


**Questions:**


1. Line 160 states how the easier tasks were further subdivided into easy, medium, and hard. How are the more algorithmically challenging tasks categorized (Line 162)? Does it make sense to use the number of valid solutions for categorizing difficulty?
2. How are the exemplars selected? How does the performance vary with the selection? Does is show high or low variance?
3. In the zero-shot case, when there is not way to ensure the proper output formatting, how do you perform automated evaluation?

**Limitations:**

The limitations discussed in the appendix are satisfactory. However, the authors should provide a reference to this discussion in the main paper.

---

> ### Author Rebuttal · Authors · 2023-08-08
>
> Thank you for acknowledging our novel contributions as well as raising valuable questions.
>
> > The performance of the model depends heavily on how the prompts are constructed. While Figure 1 shows the instructions used in the prompts clearly, to ensure reproducibility, the complete prompts for more complex strategies like COT should be provided.
>
> We will include all prompts in the appendix of the final version. Here we give a CoT example of the topological sort task:
>
> In a directed graph with 5 nodes numbered from 0 to 4:
>
> node 0 should be visited before node 4
>
> node 0 should be visited before node 1
>
> node 2 should be visited before node 1
>
> node 3 should be visited before node 2
>
> Q: Can all the nodes be visited? Give the solution.
>
> A: Since no nodes are required to be visited before node 0 and node 3, node 0 and node 3 can be visited first.
> Since visiting node 4 requires visiting node 0 first, after node 0 is visited, node 4 can be visited.
>
> Since visiting node 2 requires visiting node 3 first, after node 3 is visited, node 2 can be visited.
>
> Since visiting node 1 requires visiting node 0, node 2 first, after node 0 and node 2 are visited, node 1 can be visited. So the solution is: 0,3,4,2,1.
>
> > The motivation behind choosing the eight specific graph reasoning tasks is lacking. The ultimate goal of performing graph reasoning using LLMs is to use it to solve simple real-world tasks where graph reasoning is required. Given this motivation, the authors should provide additional justification for picking the reasoning tasks used in the paper.
>
> We discussed NLP tasks and graph tasks alignment in section A in the appendix. We will incorporate more of it in the final version given the additional page.
>
> > For the findings to apply to real-world data, the proposed prompting strategies should be tested with graphs that have real-world semantics associated with notes and edges. For example, representing nodes as numbers might allow the model to easily decouple the inherent meaning of the nodes from the meaning in the context of the given graph. This might not be an easy task if the nodes were a set of words, phrases, or actions.
>
> Thanks for your suggestion. It  is certainly a good point while our work is an initial study on this topic. Nevertheless, we provide the results of changing “node”s to “city”s, “edge”s to “roads”, and “weight”s to “distance”s in the shortest path task in Table 9.
>
> > Line 160 states how the easier tasks were further subdivided into easy, medium, and hard. How are the more algorithmically challenging tasks categorized (Line 162)? Does it make sense to use the number of valid solutions for categorizing difficulty?
>
> More algorithmically challenging tasks are divided into easy and hard subsets.
>
> > How are the exemplars selected? How does the performance vary with the selection? Does is show high or low variance?
>
> We selected exemplars according to the empirical performances on each task. The specific process is similar to the experiments in Figure 7 in the appendix. The performance varies when the exemplars are selected from different difficulties. Specifically, the performance drops when the exemplar difficulty increases.
>
> > In the zero-shot case, when there is not way to ensure the proper output formatting, how do you perform automated evaluation?
>
> Though zero-shot setting does not ensure that all the output is in the same format, there are a limited number of patterns in the LLM response. We use these patterns to automatedly evaluate. For instance, in the cycle task, “there is no cycle”, “there is a cycle” and “which creates a cycle” appear in all the output, while no output contains two or more of the three patterns, so we just match the phrases in the output to get the results. We will publicly release the LLM responses and our code to make automated evaluations upon acceptance.
>
> > The limitations discussed in the appendix are satisfactory. However, the authors should provide a reference to this discussion in the main paper.
>
> Thank you for your suggestion. We have added a reference to the limitation discussion in the revised paper.

---

> > ### Comment · Reviewer_U4cj · 2023-08-18
> > **Thank you for the clarifications**
> >
> > Thank you for answering my questions and for providing clarifications. I have no further questions.

---

### Official Review · Reviewer_E2y1 · 2023-07-06

**Soundness:** 3 good
**Presentation:** 4 excellent
**Contribution:** 4 excellent
**Rating:** 8
**Confidence:** 3

**Summary:**

This paper investigates whether large language models (LLMs) are able to solve graph algorithm problems in natural language. A benchmark NLGraph contains 29,370 problems, covering 8 graph reasoning tasks with varying complexity from simple tasks such as connectivity, cycle, and shortest path to more complex problems such as topological sort, maximum flow, bipartite graph matching, Hamilton path, and simulating graph neural networks. Various prompting techniques have been applied to evaluate the capabilities of LLMs for performing complex reasoning on graphs. Several valuable insights are provided based on the experimental results and analyses. Two prompting techniques Build-a-Graph and Algorithmic Prompting are further introduced for enhancing graph reasoning.

**Strengths:**

1. The proposed NLGraph benchmark is comprehensive, covering intuitively simple to more sophisticated tasks. Also, as the benchmark is synthetic, answers are unlikely to appear in the pretraining corpus of LLMs, making it a more robust benchmark for evaluating complex reasoning with LLMs. I think it would be a challenging and valuable testbed for evaluating the graph reasoning abilities of large language models.
2. Extensive experiments and analyses are conducted on the proposed benchmark. Specifically, results of text-davinci-003 model with various prompting techniques (e.g. Chain-of-Thoughts, Least-to-Most, Self-Consistency) are reported.  Such results and analyses help the research community to better understand the capabilities/behavior of LLMs for graph reasoning or even complex reasoning. The four key phenomenons summarized in the paper are quite counter-intuitive and thought-provoking.
3. The paper is clear and well-organized.


**Weaknesses:**

I only have several minor concerns/questions about this paper as follows:
1. Using a programming style of prompting techniques (e.g. PAL[1], PoT[2]) for solving these graph reasoning tasks is more intuitive. It could potentially address the problem of generating too many tokens of code-davinci-002.
2. Except for the results of text-davinici-003, only part of the results of code-davinci-002 are reported (i.e. cycle, shortest path, and hamilton path).  For GPT-3.5-turbo and GPT4, only 19 tasks across 8 tasks are provided. I am not sure the four key findings listed are universal for the LLMs. It would be interesting to know the overall performance of other LLMs on the benchmark. It would shed light on this problem.
3. It seems to me that including the Graph Neural Networks (GNNs) task is not well-motivated. Why do we want the LLM to perform graph convolution on a two-dimension node embedding? We already have an efficient way to calculate it even for node embeddings that have hundreds or thousands of dimensions. If the LLM can use tools (e.g. calculator, python interpreter), why do we want it to perform large number calculations itself? Similarly, LLMs can write the code of different GNNs to perform message propagation.


References:

[1]PAL: Program-aided Language Models.

[2]Program of Thoughts Prompting.

**Questions:**

See above.

**Limitations:**

The authors adequately addressed the limitations in the Appendix.

---

> ### Author Rebuttal · Authors · 2023-08-08
>
> Thank you for your helpful and constructive feedback!
>
> > Using a programming style of prompting techniques (e.g. PAL[1], PoT[2]) for solving these graph reasoning tasks is more intuitive. It could potentially address the problem of generating too many tokens of code-davinci-002.
>
> Yes, this is a very good point! Though we have considered a programming style of prompting techniques, our concern is that LLMs are trained on a large amount of text including codes for solving these graph problems, thus they could easily recite the codes to solve the problems. The evaluation results might not truly reflect reasoning ability. Anyway, this can be a good follow-up work based on the released dataset in our work.
>
> > Except for the results of text-davinici-003, only part of the results of code-davinci-002 are reported (i.e. cycle, shortest path, and hamilton path). For GPT-3.5-turbo and GPT4, only 19 tasks across 8 tasks are provided. I am not sure the four key findings listed are universal for the LLMs. It would be interesting to know the overall performance of other LLMs on the benchmark. It would shed light on this problem.
>
> It will certainly be interesting to see how more recent LLMs (e.g., GPT-4) perform on the NLGraph benchmark. However, due to monetary costs and as we will make the NLGraph benchmark publicly available, we leave it for future work.
>
> > It seems to me that including the Graph Neural Networks (GNNs) task is not well-motivated. Why do we want the LLM to perform graph convolution on a two-dimension node embedding? We already have an efficient way to calculate it even for node embeddings that have hundreds or thousands of dimensions. If the LLM can use tools (e.g. calculator, python interpreter), why do we want it to perform large number calculations itself? Similarly, LLMs can write the code of different GNNs to perform message propagation.
>
> The main purpose of the GNN task is to evaluate LLMs' graph reasoning (propagation of information) abilities instead of utilizing LLMs to perform GNN operations in real use. As the GNN task requires LMs to consider relations in the graph and perform arithmetic operations simultaneously, we consider it as a good task for evaluating graph reasoning abilities.

---

> > ### Comment · Reviewer_E2y1 · 2023-08-11
> >
> > Thanks for the clarification. I will keep my rating as accepted (8) for this paper.

---

### Official Review · Reviewer_moU8 · 2023-07-08

**Soundness:** 3 good
**Presentation:** 3 good
**Contribution:** 3 good
**Rating:** 7
**Confidence:** 3

**Summary:**

This work tried to answer the question, "Are LLMs capable of mapping textual descriptions of graphs and structures to grounded conceptual spaces and solving graph algorithm problems explicitly with natural language?" The answer to this question has profound implications for large language model applications with implicit graphs and structures, the reasoning ability of LLMs in advanced and graph-based settings, and more. This paper proposes the Natural Language Graph (NLGraph) benchmark, a comprehensive testbed of graphs and structured reasoning designed for language models and in natural language. Extensive experiments on the NLGraph benchmark demonstrate that:
1. LLMs do possess preliminary graph reasoning abilities.
2. The benefit of advanced prompting methods diminishes with complex problems.
3. Learning from examples did not happen on complex graph reasoning problems.
4. LLMs are (un)surprisingly brittle to spurious correlations in problem settings.

To improve large language models as better graph reasoners, this work proposes two instruction-based prompting approaches to better elicit the graph reasoning abilities of large language models. Build-a-Graph prompting encourages LLMs to map the textual descriptions of graphs and structures to grounded conceptual spaces before tackling the specific problem through a one-sentence instruction, while Algorithmic prompting instructs LLMs to revisit the algorithmic steps for a given task before learning from in-context exemplars.




**Strengths:**

1. The motivation of this work is to answer the question, "Are LLMs capable of mapping textual descriptions of graphs and structures to grounded conceptual spaces and solving graph algorithm problems explicitly with natural language?" is very interesting.

2. The extensive experiments are comprehensive. Experiments demonstrate that build-a-graph and algorithmic prompting successfully empower LLMs to better tackle graph reasoning problems, resulting in 3.07% to 16.85% performance gains across multiple tasks.

3. This paper provides the Natural Language Graph (NLGraph) benchmark, a comprehensive testbed of graphs and structured reasoning designed for language models and in natural language.




**Weaknesses:**

This work is mainly on testing different prompt engineering strategies and in-context learning results. If we can include more LLM fine-tuning results, it will be of great help.


**Questions:**

None

---

> ### Author Rebuttal · Authors · 2023-08-08
>
> Thank you for your time and positive feedback!
>
> > If we can include more LLM fine-tuning results, it will be of great help.
>
> It would be valuable and interesting to see LLM fine-tuning results, but it is too expensive to fine-tune LLMs such as GPT-3. As we will make the NLGraph benchmark publicly available, we leave it for future work.

---

> > ### Comment · Reviewer_moU8 · 2023-08-19
> >
> > Thanks a lot for your reply!

---

### Official Review · Reviewer_7jfe · 2023-07-15

**Soundness:** 4 excellent
**Presentation:** 4 excellent
**Contribution:** 4 excellent
**Rating:** 8
**Confidence:** 4

**Summary:**

This paper evaluates and studies the performance of state-of-the-art LLMs on graph reasoning-based tasks. For this, the authors construct a testbed of graph and structured reasoning tasks, comprising of 29K problems on 8 graph reasoning tasks such as topological sorting and max flow. Each task has three subsets based on problem difficulty and the default metric is exact match accuracy. The comprehensive evaluation conducted by the authors shows that LLMs perform well on graph reasoning tasks but advanced-prompting algorithms such as CoT, L2M, Self-Consistency based COT, etc. do not yield performance improvements on complex tasks. The authors also show that the LLMs evaluated (mainly text-davinci-003, but also others from OpenAI) are brittle to spurious correlations in the problem settings. Finally, the authors propose two very simple, domain-specific prompting methods, named Build-a-Graph and Algorithmic Prompting, which improves upon zero-shot performance of text-davcinci-003 on the tasks.

**Strengths:**

1. Comprehensive and Interesting: I really like this work. This is primarily an evaluation paper, and the authors do a great job of conducting a systematic and comprehensive study of LLM performance on graph reasoning tasks.
2. A Useful Resource for the Community: A number of reasoning problems could be reduced to implicit reasoning over graphs and the proposed benchmark, NLGraph will help the research community evaluate LLMs on such reasoning problems.

**Weaknesses:**

1. Lack of Systematic Comparisons between LLMs: Although I am empathetic to the plight of researchers conducting research on black-box models, only through an API, I feel the work would have been more valuable had the researchers also focused on a comparison between 3 generations of LLMs: GPT-3, GPT-3.5 and GPT-4. It would have enumerated a few things: the impact of instruction finetuning since the chat models are more aggressively instruction finetuned, the gains in performance across these models and the impact of advanced prompting with scale/model generation. The qualitative analysis in Appendix E on 19 problems is insufficient in ascertaining any of the performance jumps.


**Questions:**

1. From Appendix E, it appears that GPT-4 is considerably more capable than GPT-3.4-Turbo. A number of other works have found GPT-4 to show considerably higher performance, thereby hitting the thresholds of practical utility. Can you comment on any of this behavior wrt the graph reasoning tasks?
2. I am surprised that zero-shot is the best performing method in Figure 2 (right, max-flow task). Similarly, I am finding it hard to understand why CoT underperforms zero-shot. How much time was spent iterating on CoT? Can you comment of this exact trend holding on for GPT-4 as well?
3. To arrive at the Build-a-Graph and Algorithmic prompting methods, how many negative prompts did you iterate through? It is clear that domain specific contextual knowledge is being leveraged by LLMs to solve this task, but I am interested in the meta-question of how did you arrive these methods?

**Limitations:**

Yes, the authors have adequately addressed the limitations.

---

> ### Author Rebuttal · Authors · 2023-08-08
>
> Thank you for the positive feedback and helpful suggestions!
>
> > Lack of Systematic Comparisons between LLMs
>
> We believe it is interesting to see how more recent LMs (e.g., GPT-4) perform on the NLGraph benchmark. However, due to the budget we had and as we will make the NLGraph benchmark publicly available, we leave it for future work.
>
> > From Appendix E, it appears that GPT-4 is considerably more capable than GPT-3.4-Turbo. A number of other works have found GPT-4 to show considerably higher performance, thereby hitting the thresholds of practical utility. Can you comment on any of this behavior wrt the graph reasoning tasks?
>
> According to the experiments we did, GPT-4 is indeed more capable than GPT-3.5-turbo. However, it still fails in some easy cases such as case 3 in Table 11, which humans can easily solve, and in many difficult cases, indicating it only possesses preliminary graph reasoning abilities.
>
> > I am surprised that zero-shot is the best performing method in Figure 2 (right, max-flow task). Similarly, I am finding it hard to understand why CoT underperforms zero-shot. How much time was spent iterating on CoT? Can you comment of this exact trend holding on for GPT-4 as well?
>
> We have tried several versions of CoT on a small subset of the problems and finally choose the form that performs the best.
>
> Problems such as max-flow are rather difficult, even for humans. A human struggles to solve a max-flow task with about ten nodes, even when some exemplars are given.  Due to the difficulty, LLMs fail to learn the correct way to generate intermediate steps or learn from in-context exemplars.The extra text in CoT might be hard for the model to grasp and does not provide a positive impact. We believe the trend still holds on for GPT-4, though it needs experiments to finally prove this.
>
> > To arrive at the Build-a-Graph and Algorithmic prompting methods, how many negative prompts did you iterate through? It is clear that domain specific contextual knowledge is being leveraged by LLMs to solve this task, but I am interested in the meta-question of how did you arrive these methods?
>
> We first come up with a set of prompting methods that intuitively help solve graph reasoning problems, and then test them.  Build-a-Graph and Algorithmic prompting methods work the best. Other methods such as asking LMs to answer some simple questions about the graph property before answering the true question do not work well. We also try variants of our prompting methods and provide the results in Table 7 in the appendix.

---

### Author Rebuttal · Authors · 2023-08-08

We would like to express our sincere gratitude to the reviewers for providing us with valuable feedback. These constructive comments have been instrumental in improving the quality of our work. We are glad that our efforts have been well-received by the reviewers, and we are confident that their insightful feedback will further enhance the impact of our paper. With their guidance, we have outlined clear plans to incorporate their suggestions in the camera-ready version of the paper, and we are excited to present an even stronger final version.

---

> ### Comment · Reviewer_EhYM · 2023-08-18
>
> Acknowledging that I've read the response.

---

### Decision · Program_Chairs · 2023-09-21

**Decision:**

Accept (spotlight)

**Comment:**

This paper presents a comprehensive evaluation of Large Language Models (LLMs) on graph reasoning tasks. The authors construct a testbed of graph and structured reasoning tasks, NLGraph (29,370 problems, covering eight graph reasoning tasks), and evaluate the performance of LLMs on these tasks. The reviewers commend the authors for their systematic and comprehensive study, and agree that the proposed benchmark will be a valuable resource for the research community. The reviewers also appreciate the authors' efforts in proposing two simple, domain-specific prompting methods, Build-a-Graph and Algorithmic Prompting, which improve upon zero-shot performance of LLMs on the tasks.

However, the reviewers also pointed out some areas for improvement. One reviewer suggested that the work would have been more valuable if the authors had also focused on a comparison between different generations of LLMs. Also, one reviewer questioned the inclusion of the Graph Neural Networks (GNNs) task and suggested using a programming style of prompting techniques for solving these graph reasoning tasks.

Despite these minor concerns, the reviewers agree that the paper is technically strong, with novel ideas on at least one area.